# FROM MODELS TO MICROTHEORIES: DISTILLING A MODEL'S TOPICAL KNOWLEDGE FOR GROUNDED QUESTION ANSWERING

**Nathaniel Weir**[†,*]**, Bhavana Dalvi Mishra**[♠]**, Orion Weller**[†]**, Oyvind Tafjord**[♠]**,
Samuel Hornstein**[#]**, Alexander Sabol**[#]**, Peter Jansen**[♠,‡]**,
Benjamin Van Durme**[†]**, Peter Clark**[♠]

[†]Johns Hopkins University    [♠]Allen Institute for AI
[#]Thomas Jefferson University    [‡]University of Arizona
nweir@jhu.edu, peterc@allenai.org

## ABSTRACT

Recent reasoning methods (e.g., chain-of-thought) help users understand how language models (LMs) answer a single question, but they do little to reveal the LM's *overall understanding*, or "theory," about the question's *topic*, making it still hard to trust the model. Our goal is to materialize such theories - here called *microtheories* (a linguistic analog of logical microtheories (Blair et al., 1992)) - as a set of sentences encapsulating an LM's core knowledge about a topic. These statements systematically work together to entail answers to a *set* of questions to both engender trust and improve performance. Our approach is to first populate a knowledge store with (model-generated) sentences that entail answers to training questions, and then distill those down to a core microtheory which is concise, general, and non-redundant. We show that, when added to a general corpus (e.g., Wikipedia), microtheories can supply critical information not necessarily present in the corpus, improving both a model's ability to ground its answers to verifiable knowledge (i.e., show how answers are systematically entailed by documents in the corpus, grounding up to +8% more answers), and the accuracy of those grounded answers (up to +8% absolute). We also show that, in a human evaluation in the medical domain, our distilled microtheories contain a significantly higher concentration of topically critical facts than the non-distilled knowledge store. Finally, we show we can quantify the coverage of a microtheory for a topic (characterized by a dataset) using a notion of *p-relevance*. Together, these suggest that microtheories are an efficient distillation of an LM's topic-relevant knowledge, that they can usefully augment existing corpora, and can provide both performance gains and an interpretable, verifiable window into the model's knowledge of a topic.[1]

## 1 INTRODUCTION

What do language models (LMs) know about the topics they converse about? Despite their success, LMs are opaque - users do not have good visibility into the LM's knowledge, creating challenges for engendering trust in the model. Recent work has tried to address this for answers to a single question by having the LM provide various explanations and chains of thought that support an answer, e.g., (Danilevsky et al., 2020; Wei et al., 2022). However, such expositions only reveal snippets of the LM's latent knowledge and do little to convey the model's overall understanding, or "theory", about the topic at hand, limiting a user's confidence in the model's behavior. Conversely, if we had a method that could distill a model's overall theories into an inspectable, verifiable form, this could help users see how answers follow from those theories.

Our goal is to distill such theories - which we refer to as *microtheories* (Mts) (a linguistic analog of logical microtheories (Blair et al., 1992)) - as a set of sentences encapsulating an LM's core knowledge about a *topic*. Our approach uses a question-driven methodology, where a set of questions characterizes a topic (i.e., a topic is defined as a distribution over questions). We first prompt the

---

*Work done in part during internship at Allen Institute for AI

[1]Code for this work can be found at https://github.com/nweir127/microtheories/.

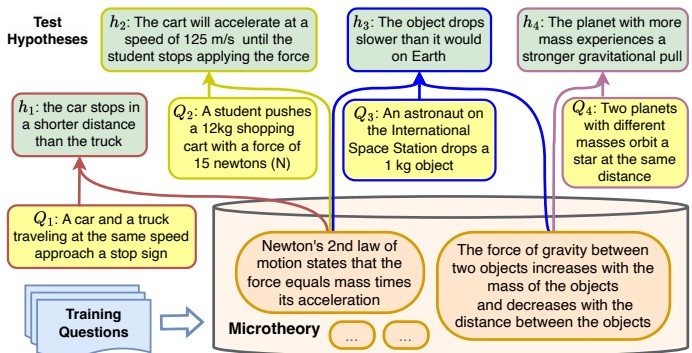

Figure 1: Given a set of topical training questions, we construct a microtheory, a set of statements articulating a model's core, reusable knowledge about that topic. These help prove (entail) answers to test questions.

LM to enumerate facts that entail answers to training questions, then identify the core, abstractable pieces of knowledge frequently reused for many questions in that set, then distill them down to a core microtheory that is concise, general, and non-redundant. The microtheory is intended to contain the core principles about the topic (e.g., Newton's laws of motion, for physics), extracted from the LM, that can help derive corpus-grounded answers to topical questions.

To answer questions with the help of a microtheory, we use the well-known process of *textual entailment* (Dagan et al., 2005; Lai et al., 2017). Textual entailment is a linguistic inference process that identifies when one (or more) statement(s) "reasonably implies" another (Magnini, 2015), producing a (potentially multistep) entailment tree, analogous to a proof tree (Dalvi et al., 2021). While entailment lacks a fully formal definition (modern entailment engines are instead trained on examples), it provides a well-defined mechanism for deciding if, and how, an answer systematically follows from a set of statements. This allows us to both use and evaluate microtheories for question-answering.

We evaluate our microtheory approach in two domains using two existing datasets: **K-12 science**, using the ARC dataset (Clark et al., 2018), and **medicine**, using the MedQA dataset (Jin et al., 2021). We cluster each dataset into topics, construct microtheories on a training split of each topic, and test on the remainder. We find that when added to a general corpus (here, Wikipedia), microtheories can supply critical, topical information not necessarily present in the corpus, improving both the model's ability to ground its answers (i.e., show how those answers are entailed by the corpus, fully grounding up to +8% more answers), and the accuracy of those grounded answers (up to +8% absolute). We also show that, in a human evaluation in the medical domain, our distilled microtheories contain a significantly higher concentration of topically critical facts than in the non-distilled knowledge store. Finally, we propose a metric, *p-relevance*, to quantify the completeness of a microtheory. By correlating $p$-relevance with training set size, we can predict how many training questions will be needed to generate a Mt relevant to some $p$ percent of test questions. In summary, our contributions are:

• A method for distilling microtheories from a model, given a QA dataset, that articulates the model's core knowledge about the dataset topic. This is the first method to try to distill a model's overall understanding of a topic in this way.

• Evaluations that show microtheories can improve both a model's ability to ground its answers to verifiable facts in a corpus, and the accuracy of those answers.

• A human evaluation in the medical domain, showing our method distills out topically critical facts from the general pool.

• A new metric, *p-relevance*, to quantify the completeness of a microtheory for a dataset.

Together, these suggest that microtheries are an efficient distillation of an LM's topic-relevant knowledge, that they can usefully augment existing corpora, and can provide both performance gains and an interpretable, verifiable window into the model's knowledge of a topic.

## 2 RELATED WORK

The notion of a microtheory - a small, coherent, topical representation of knowledge - arose in formal logic to help organize a large, logical KB into useful, topical subsets, each sharing the same underlying assumptions (context) (Lenat et al., 1990; Blair et al., 1992). A logical microtheory

aimed to model a specific aspect of reality as a self-contained, logical system, akin to a piece of "computational clockwork" (Clark et al., 2004). Our work here can be seen as a modern approach to this, similarly aiming to capture an understanding of a topic but where the units of knowledge are in NL and reasoning is via textual entailment.

It has long been recognized that LMs contain substantial knowledge, and can even be viewed as a new kind of "knowledge base" (Petroni et al., 2019; Bosselut et al., 2019). Some work has sought to materialize some of the model's factual knowledge either as knowledge graph triples (Alivanistos et al., 2022; Hao et al., 2022), or NL statements, e.g., BeliefBank (Kassner et al., 2021). Other work has generated "belief graphs" showing how model-believed facts entail answers (Kassner et al., 2023; Tafjord et al., 2022; Jung et al., 2022). However, while providing some visibility, these methods only provide disconnected slices of the model's latent knowledge, at best related to a *single question*. Our work takes this further, aiming to extract a model's overall understanding of a *topic*.

One notable manual attempt at a topical, linguistic microtheory is WorldTree, a corpus of several thousand statements (aiming to) cover all the knowledge required for a syllabus in elementary science (Xie et al., 2020), constructed at significant expense. Our goal with microtheories can be seen as automating such a construction process, and extracting such topical knowledge from language models directly (we later compare our Mts against WorldTree in §5.1 and §5.2).

More broadly, there has been substantial work in generating explanations with language models, to help users see the rationale behind a model's answers (Wiegreffe & Marasovic, 2021; Danilevsky et al., 2020; Wei et al., 2022). However, much of this work is again question-centric, and sometimes lacks a clear definition of whether an explanation adequately "supports" an answer. To more clearly define the relationship between supporting evidence and an answer, and for linguistic reasoning in general, there has been a resurgence of interest in *textual entailment* as a linguistic inference process, with numerous, high-quality, multi-step entailment engines becoming available, e.g., SCSearch (Bostrom et al., 2022), NLProofS (Yang et al., 2022), Entailer (Tafjord et al., 2022), IRGR (Ribeiro et al., 2022), TreeWise (Weir et al., 2024), and MetGen (Hong et al., 2022). In our work, we leverage this development to clearly define whether a microtheory supports an answer or not.

Finally, our work is distinct from work on library learning, which seeks to induce generalizations over data, e.g., (Muggleton, 1991; Wang et al., 2023). Rather, our goal here is to articulate generalizations that the LM appears to have already formed.

## 3 APPROACH

### 3.1 BACKGROUND: TEXTUAL ENTAILMENT

To make meaningful claims about microtheories, we need a clear definition of what it means for an answer to follow from a microtheory (or a corpus in general). For this, we use the well-known notion of *textual entailment* in NLP: a linguistic inference process that identifies whether one or more statements "reasonably imply" another (Dagan et al., 2005). Modern entailment engines perform a soft version of theorem proving by searching for *entailment trees*, analogous to proof trees, showing how a query hypothesis $H$ follows from statements in a theory $\mathcal{T}$, where both elements are stated in NL. Here, H is a sentence (commonly a declarative answer to a question $Q$), $\mathcal{T} = [t_1, \ldots t_n]$ is a corpus of text passages, e.g. sentences or documents, and an entailment tree $T$ is a tree whose root is $H$, whose leaves $L = [l_1, \ldots, l_l]$ are a subset of texts in $T$, and whose individual steps have the form $[p_1, \ldots p_j] \models_Q h_k;\ j \in \{1, \ldots, |T| - 1\}$, where $\models_Q$ denotes atomic textual entailment given the context of question $Q$ ($\models_Q$ is typically learned from examples). Multiple alternative trees $T_i$ may be found supporting the same hypothesis $H$, analogous to finding multiple proofs for a hypothesis in formal logic. For large $\mathcal{T}$s, the search space can be very big. To handle this, entailment engines such as IRGR (Ribeiro et al., 2022) or TREEWISE (Weir et al., 2024) use iterative text retrieval to look up relevant statements and use them for grounding and decomposing hypotheses.

**Definitions:** If a tree $T_i$ shows how $H$ is entailed from a set of corpus texts $L_i$, we say $H$ is **grounded** in the corpus. We also say the leaf statements $L_i$ provide an **argumentative basis** for accepting $H$. If there are multiple trees with different leafsets, we say that $L_1, L_2, \ldots$ constitute **alternative bases** for supporting $H$.

### 3.2 CONSTRUCTING MICROTHEORIES

We now describe a method to automatically construct a microtheory, namely a list of model-generated NL assertions that are "most important" for entailing answers to questions about a given topic, characterized by a set of training questions. Microtheories are intended to capture the core knowledge about the topic, and we evaluate different operationalizations of "most important"

shortly. First, for a given training set $[Q_1, \ldots Q_d]$, we extract from an LM, via prompting, sets of facts that it would use to support the correct answer to each question (§3.2.1). Then, to reduce redundancies and promote generalizability, we apply a series of optimizations to distill the core kernels of knowledge by identfying entailment relations (§3.2.2). Finally, we propose (and later evaluate) three alternative ways of optimizing the microtheory contents given a particular size budget (§3.2.3).

### 3.2.1 RAW FACT POOL EXTRACTION

We first prompt an LM to regurgitate facts on a question-by-question basis in order to populate an initial factpool $\mathcal{F}$. We use a few-shot learning prompt that cues the LM to perform a version of 'chain-of-thought' reasoning (Wei et al., 2022) catered towards extracting facts. Each example in the prompt contains a question $Q_e$, answer options to $Q_e$ in the form of hypotheses $h_1, \ldots h_n$, a list of supporting statements for $Q_e$ (a question-specific 'theory'), and a list of entailment steps that compose a subset of statements into complex inferences that answer the question by supporting the correct $h$. Prompting details for this GenFacts() method can be found in §C. As training examples, we use EntailmentBank (Dalvi et al., 2021), a dataset of science questions paired with lists of supporting facts that combine into an entailment tree for the correct answers. We define $\mathcal{F}$ as:

$$\mathcal{F} = \bigcup_1^d \text{KeepIfGeneric}(\text{GenFacts}(Q_i, [h_1, \ldots, h_n])) \tag{1}$$

Because the LLM often generates context-specific sentences via GenFacts() that we wouldn't want to keep in our microtheory, e.g. about specific entities in the question (e.g. "the man is running"), we create a filter (KeepIfGeneric() in (1)) that prompts the LLM to classify each sentence as generic or context-specific and keeps only the generics.

### 3.2.2 SBERT AND ENTAILMENT-BASED CONDENSATION

As fact lists are extracted independently for each question, there is substantial redundancy within the unified pool. We use a pair of techniques to remove statements and produce condensed pool $\mathcal{C} \subseteq \mathcal{F}$:

**Soft Deduplication.** We identify paraphrases using a Sentence Transformer (SBERT; Reimers & Gurevych, 2019) trained to maximize the embedding cosine similarity between same-meaning facts. We perform a linear pass of $\mathcal{F}$ and remove every fact $f_i$ for which $\exists f_j \in \mathcal{F}, j > i, \cos(\text{SBERT}(f_i), \text{SBERT}(f_j)) > t$ for some threshold $t$.

**Entailment Condensation.** We perform a second pass through $\mathcal{F}$ to find pairs $f_i, f_j$ s.t. $f_i \models f_j$, and remove $f_j$. We use a fine-tuned neural cross-encoder to predict $\models$. As it would be infeasible to check all $|\mathcal{F}|^2$ pairs of facts, we leverage SBERT to find the most promising candidates. We check for entailment between any two facts $f_i, f_j$ iff $\cos(\text{SBERT}(f_i), \text{SBERT}(f_j)) > u, u < t$.[2]

### 3.2.3 ENGINE USAGE-BASED OPTIMIZATION

After removing duplicates and entailments, we still see facts that serve the same *functional* roles in many questions but imply slightly different things, e.g. for physics:

   (A) The force required to cause a given acceleration is determined by the mass of an object
   (B) The force acting on an object equals mass times acceleration

While this might be acceptable for a knowledge store with infinite storage, we instead seek a more concise representation without redundancies. Towards this goal, we consider three alternative approaches, illustrated in Figure 2, to identify some $n$ most effective statements to retain in the Mt. We will refer to the resulting sets as $n$-**Mts**.[3] Under this budget, we would not want to retain both (A) and (B) in the $n$-Mt if it meant discarding some (C) that is important to other questions.

**1. Top-$n$ Most Used Facts (Mt$_{\text{usage}}$).** We seek the set of facts that best "covers" the argumentative bases for a given dataset of hypotheses. One intuitive way to choose these facts is to feed them to an entailment engine and see which ones are used most frequently to explain hypotheses in the dataset. We refer to this approach as the *usage* $n$-Mt. Using questions $Q_1 \ldots Q_d$ and hypotheses $h_1 \ldots h_d$:

$$n\text{-Mt}_{\text{usage}} = \underset{\mathcal{M} \subseteq \mathcal{C}, |\mathcal{M}| = n}{\arg\max} \sum_{f_i \in \mathcal{M}} \sum_{j=1}^d \delta(f_i \in \text{LEAVES}(\text{ENGINE}(h_j, \mathcal{C}; Q_j))) \tag{2}$$

---

[2] We use `all-mpnet-base-v2` for $\text{SBERT}(\cdot)$, `sileod/deberta-v3-large-tasksource-nli` for $\models$, and set $t = .9$ and $u = .3$.

[3] $n$ is a hyperparameter; below, we observe that $n$ represents a trade-off between concision and performance.

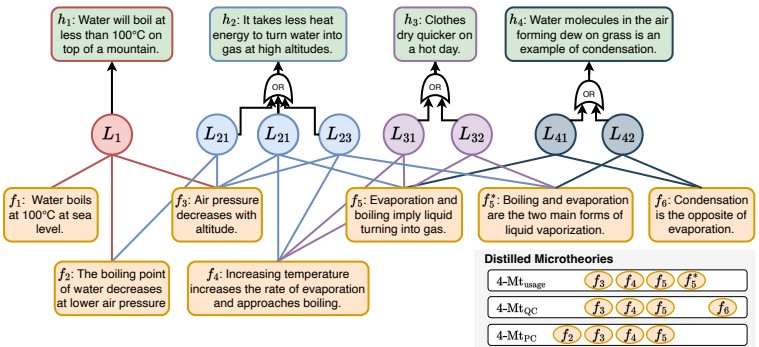

Figure 2: Comparison of distillation techniques for extracting $n$-Mts from a larger fact pool $\mathcal{C}$ based on their entailments ($L$s of different training hypotheses ($h$s)). The *usage* approach prioritizes facts based on the number of hypotheses for which they are used. This risks keeping facts that serve the same role in explaining the same $h$s (e.g., $f_5$ and $f_5^*$ are both kept, even though only one is needed to explain $h_2$, $h_3$, and $h_4$). The *question coverage* (QC) approach maximizes the number of $h$s for which the $n$-Mt contains *all* of a supporting argument's leaves (here $h_2$, $h_3$, and $h_4$). This risks failing to cover some $h$s at all (e.g. $h_1$). *Partial coverage* (PC) maximizes the total *fraction* of the argument covered for all questions (preferring the most covered argument for each $h$, if there is more than one). In this example, coverage = .66+1+1+.5 = 3.16 (of a possible 4).

where ENGINE returns a (possibly empty) set of entailment trees proving $h_j$ from fact pool $\mathcal{C}$, and LEAVES refers to the set of all facts used at least once in any of those trees. This objective does *not* address the functional redundancy issue for fact pairs such as (A) and (B). If (A) and (B) serve the same role in entailing hypotheses, then both facts will be used for a similar number of questions; if one is in the top-$n$ most used facts, the other will likely also be.

**2. Maximum Question Coverage (Mt$_{QC}$).** We use an objective function minimizing the facts in a theory while maximizing the number of train hypotheses "covered." For each question $Q_j$ with bases $\mathcal{L}_j = L_{1j}, L_{2j}, \ldots$ (returned by LEAFSETS), we try to fully cover at least one $L_{kj}$ for as many questions as possible using $\leq n$ facts. We use a linear program described in §A.1.

$$n\text{-Mt}_{QC} = \underset{\mathcal{M} \subseteq \mathcal{C}, |\mathcal{M}|=n}{\arg\max} \sum_{j=1}^{d} \delta(\exists L_{kj} \in \text{LEAFSETS}(\text{ENGINE}(h_j, \mathcal{C}; Q_j)); L_{kj} \subseteq \mathcal{M}) \quad (3)$$

**3. Maximum Partial Coverage (Mt$_{PC}$).** Equation (3) poses question coverage as a binary variable: for each $Q_i$, either an entire basis is kept, or $Q_i$ is not covered. We consider a variant that loosens this constraint, modeling the *partial* coverage of questions.[4] We use an objective function that maximizes the per-question partial basis coverage (LP described in §A.2). Given each question $Q_i$ associated with bases $L_{1i}, L_{2i}, \ldots$, we try to best cover some $L_{ki}^*$.

$$n\text{-Mt}_{PC} = \underset{\mathcal{M} \subseteq \mathcal{C}, |\mathcal{M}|=n}{\arg\max} \sum_{j=1}^{d} \underset{L_{kj} \in \text{LEAFSETS}(\text{ENGINE}(h_j, \mathcal{C}; Q_j))}{\arg\max} \frac{|\mathcal{M} \cap L_{kj}|}{|L_{kj}|} \quad (4)$$

## 4 EXPERIMENTS: CONSTRUCTING MICROTHEORIES

We apply our method to construct microtheories for two domains: **ARC** (Clark et al., 2018), comprising grade-school science questions, and **MedQA** (Jin et al., 2021), containing USMLE medical school exam questions. Our goal is to extract generalizable knowledge statements using training questions that can be readily applied to solve test questions. This requires train/test splits that are about the same topics. To do this, we create mini-splits of each dataset containing topically similar questions with 6-900 training questions using a hierarchical topic clustering algorithm we describe in §B. For ARC, we make a split with 9 topic clusters including "Genetics and Evolution", "Force, Mass, Acceleration and Gravity", and "Plant Growth and Reproduction." For MedQA, we make a split with 4 clusters including "The Effects of Smoking on Health" and "Kidney Function and Disorders." (See Figure 3 for details).

### 4.1 MICROTHERY CONSTRUCTION DETAILS

To generate the initial fact pools (§3.2.1), we use GPT-4 (`gpt4-0613`). For entailment reasoning, we use TREEWISE (Weir et al., 2024), a recent, off-the-shelf, search-based SOTA entailment engine

---

[4]For 3 questions, it might be better to cover 70%, 70%, & 70% of a basis for each than 100%, 100%, & 0%.

|  | **ARC** | **MedQA** |
|---|---|---|
| **Dataset Details** | | |
| # Questions | 854/127/249 | 641/94/186 |
| # Topic Clusters | 9 | 4 |
| Ext. Corpus | Wikipedia | Wiki+Textbooks |
| **Training Extraction Results** | | |
| $|\mathcal{F}|$ | 20,561 | 16,722 |
| $|\mathcal{C}|$ | 8,722 | 10,908 |
| # Qs with Proof | 744 (89%) | 562 (88%) |
| Min #Fs to Cover | 800 | 890 |
| Fact/Q Ratio | 1.08 | 1.58 |

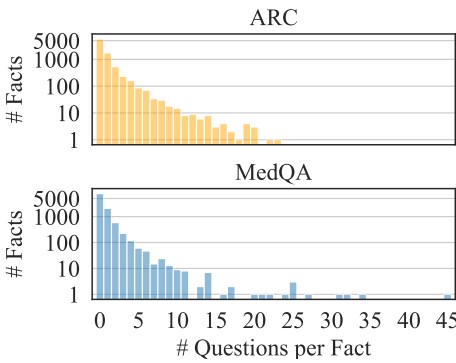

Figure 3: (Upper) Dataset details for the two domains. (Lower) Results of microtheory extraction from training data. The last two rows result from the "min-fact" LP.

Figure 4: Histogram of training question per fact in $\mathcal{C}$ and 650-850 training questions. ARC facts are used more frequently than MedQA.

that contains fine-tuned models to verify individual entailment steps (we use its tuned ChatGPT). To demonstrate our work is not closed-model dependent, we show MedQA results using Mixtral-8x22B-Instruct-v0.1 (Jiang et al., 2024) in §G. We construct $n$-Mts for $n$=25 to 1000 using the *usage*, *question coverage* and *partial coverage* methods described in §3.2. Details are found in §D.

Figure 3 displays statistics about the number of questions and size of the resulting fact pools created during extraction. It shows that the redundancy reduction techniques in §3.2.2 remove around 55% of ARC facts and 25% of MedQA. Figure 4 shows histograms for how frequently each fact in $\mathcal{C}$ was used in the 650-850 questions. In both datasets, a substantial fraction of facts (5000/8000) are never used in a proof. A few dozen facts from the ARC pool are used in more than 10 questions; fewer from the MedQA pool are used this frequently, suggesting that MedQA questions target more diverse pieces of knowledge than ARC, though a handful of MedQA facts are used in $\geq$30 questions.

To investigate this difference further we implemented a "min-fact" linear program that found the smallest set of facts that fully covers a basis for every hypothesis the prover was able to ground. Figure 3 shows that the 744 ARC training questions provable from $\mathcal{C}$ required a minimum of 800 facts for "full coverage," a ratio of 1.08 facts per question, while MedQA required nearly twice that rate. This highlights that the total amount of knowledge statements required to answer MedQA questions is substantially higher, which is unsurprising given the domains – grade school- versus medical school-level topics. We refer readers to §E for illustrations of the fact selection processes by Mt$_{PC}$ and Mt$_{QC}$, examples of microtheory fact lists, and further qualitative analysis.

### 4.2 Coverage of Training Hypotheses

How well do the usage-based optimization methods cover argumentative bases for the training hypotheses? Figure 5 shows the fraction of full and partial coverage by each Mt approach.[5] First, as expected, coverage increases with the size of the Mt. Second, we see that our third optimization method, partial coverage (PC), performs best here, with Mt$_{PC}$ (orange bars) consistently obtaining a higher coverage than either Mt$_{QC}$ or Mt$_{usage}$ (green, blue). Finally, we observe full coverage is generally lower on MedQA than ARC, suggesting that the MedQA domain contains fewer reusable principles and more idiosyncratic facts than ARC. We later explore this conjecture further §5.3.

## 5 Evaluation

Our goal is that microtheories contain the core, reusable principles that underlie a topic. Note that a microtheory (or any theory, for that matter) cannot contain *all* the knowledge required to answer *all* topical questions, given that an infinite long-tail of idiosyncratic facts is also required in addition to core principles. For example, to determine that a dropped pencil (say) will fall to the ground requires not just general knowledge of gravity, but also the idiosyncratic fact that a pencil is a denser-than-air object. To supply such facts, we assess our microtheories when combined with a large, general corpus (Wikipedia and textbooks, see §5.1), treating each source as a flat retrieval index over knowledge statements. We evaluate along three dimensions:

1. **Grounding (§5.1):** For the *correct* answers to the test questions, do microtheories improve the number of answers that can be fully grounded in the (combined) corpus? If so, this would suggest

---

[5]We define fractional coverage of a question $Q_j$ as in §3.2.3: the highest fraction of leaves amongst all argumentative bases $L_{kj}$ found by the engine for the question.

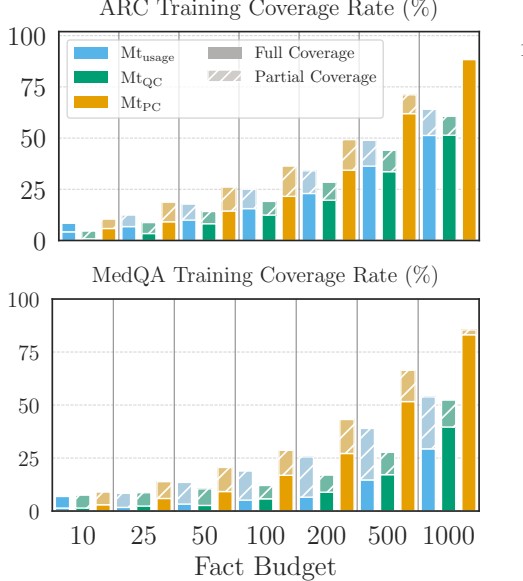
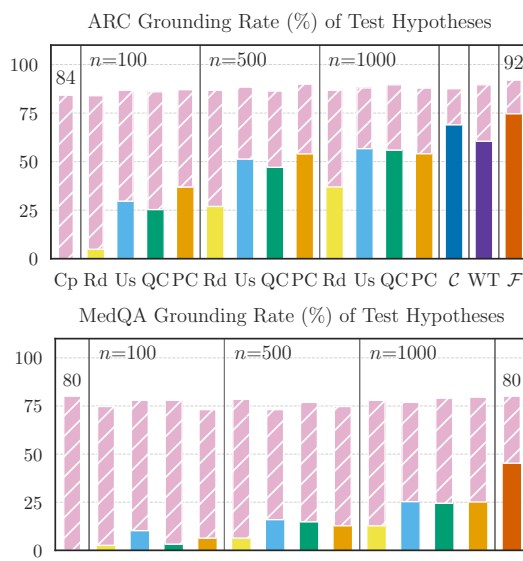

Figure 5: Rate at which different microtheory approaches entail training hypotheses. Lighter bars are the sum of fractional coverages for questions not fully covered by the Mt. Total coverage for ARC questions is generally higher than MedQA. The 1000-Mt$_{PC}$s effectively fully cover the training hypotheses for which a basis was found in both domains; since some hypotheses did not have a basis found, 100% coverage is impossible.

Figure 6: % of test hypotheses fully groundable after adding various microtheories ($n$-sized **R**an**d**om, **Us**age, **QC**, & **PC** Mts) to a base **C**or**p**us. We show the fraction of leaves grounded in the Mt (solid) and the base corpus (striped). We see **optimized Mts outperform the random baseline** (Rd), and trends show the tradeoff of conciseness vs. performance. $\mathcal{F}$ **provides up to +8% grounding** over using just the corpus and exceeds the benefit of adding the handcrafted **W**orld**T**ree corpus.

that microtheories are supplying important, topical information not necessarily present in the original corpus. We also measure what proportion of those groundings are supplied by the microtheory vs. the original corpus, and the impact of Mt size.

2. **Question Answering (§5.2):** Using the entailment engine to answer the questions (i.e., selecting the entailed answer option, or the option with the highest-scoring entailment tree if more than one), do microtheories improve QA accuracy? If so, this suggests Mts supply important information to help a downstream task. We evaluate this in our primary (science) domain, ARC.

3. **Relevance (§5.3):** To complement our empirical evaluations, we also perform a rigorous, (human) expert evaluation of microtheory statements to assess, from a human perspective, whether they appear to be critical, core domain principles (rather than idiosyncractic facts).

## 5.1  GROUNDING

We first evaluate, for the *correct* answers to test questions, whether microtheories improve the number of answers that can be fully grounded by the entailment engine (i.e., can be "proved" via entailment) when added to a base, general corpus. For our primary domain (ARC), we use Wikipedia as our base corpus, and for MedQA, we use a combination of Wikipedia and the set of medical textbooks released by Jin et al. (2021). These corpora are vastly larger than the size of the microtheories.

**Baselines.** We compare the $n$-Mt methods against a baseline that samples $n$ random facts from $\mathcal{F}$. We also compare against using the entire pool $\mathcal{F}$. For ARC, we also compare against the WorldTree resource, the version of which we use has 12,657 facts. We also consider using only the WorldTree facts that its annotators labeled as "core" for at least one question in their dataset (69% of facts).

**Test Grounding Results.** Figure 6 shows the hypothesis grounding rates when various microtheories are added to the respective base corpora (Wikipedia, MedQA textbooks). A hypothesis is considered "grounded" if the engine found at least one entailment tree for it rooted fully in the Mt, corpus, and/or context. Each bar is broken down into the portions of trees grounded to the Mt (solid) vs. the base corpus (striped).[6] In both domains, comparing the first bar (Cp, base corpus only) and last bar ($\mathcal{F}$, using the full fact pool as a microtheory), we see grounding rates are substantially

---

[6]If the prover finds multiple trees, we take the one with the highest fraction of microtheory-rooted leaves.

higher, suggesting that **microtheories can improve the the model's ability to ground its answers to verifiable knowledge**. This capability is important, as it allows models to show how answers systematically follow from inspectable sources.

For ARC, the engine grounds up to 8% more hypotheses (92% vs. 84%), and uses facts from $\mathcal{F}$ at a rate of 75% (solid part of the right-hand bar, i.e., 3/4 of the required entailment knowledge comes from the Mt). In MedQA, the hypotheses are grounded at a similar rate, but 45% of the knowledge comes from the Mt instead of the corpus. For ARC, we see a clear trend that as microtheory size (budget) increases, the engine can ground more answers (overall bar heights), and ground to more of the Mt knowledge (solid bars). For MedQA, the pattern is weaker: while larger Mts provide more grounding knowledge (solid bars), there is no clear trend in overall grounding rates, suggesting that in this domain, there may be fewer core, general principles vs. a large number of idiosyncratic facts. We examine this conjecture further in §5.3.

For ARC, we compare against a notable, manual attempt at a microtheory called WorldTree, a corpus of several thousand statements (aiming to) cover all the knowledge required for the ARC syllabus (Xie et al., 2020), constructed at significant expense (Figure 6, bar labeled WT). We see that **the n=1000 Mts result in similar rates of overall grounding and grounding to the Mt yet were built automatically and are less than 10% the size of WorldTree.** Thus, our approach can be seen as automating this previously expensive process of constructing such resources.

## 5.2 QUESTION ANSWERING

Figure 7 shows the results of running multiple-choice question answering on the 9 ARC topics. For each question, the answer option that the entailment engine can ground in ("prove from") the corpus is selected, or the option with the highest scoring entailment tree is selected if multiple options are entailed. Using only Wikipedia as the knowledge source (Corpus, leftmost bar) achieves 69.2% QA accuracy. Adding 100 or 500 facts with any approach generally does not improve on this and can decrease it using 100 random facts.[7] When adding 1000-fact theories, all distillation methods add 4 points in QA (73%). Adding the entire fact pool $\mathcal{F}$ further improves performance by 5 points (78%). These results suggest the following:

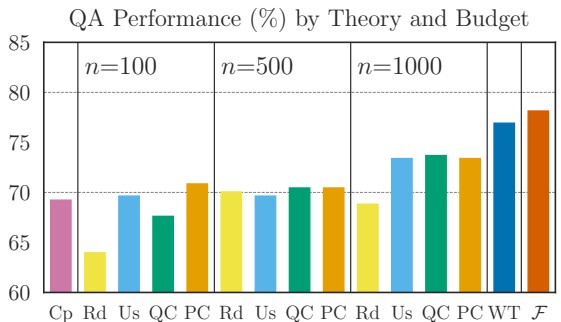

Figure 7: Question Answering performance on ARC topical test questions using Wikipedia plus various Mts (**R**and**om**, **Us**age, **QC**, **PC**) as knowledge sources. $\mathcal{F}$ adds nearly 10% over using only the base **Cor**p**us** of Wikipedia despite being a tiny fraction of its size. This exceeds the benefits of the handwritten **W**orld**T**ree corpus. ($n$=1000)-Mts improve QA by 4%. This suggests that they contain important task information that is either not in Wikipedia or is hard to find.

• Adding Mts to Wikipedia shows a benefit to QA, despite their being orders of magnitude smaller.

• Mt size correlates with QA accuracy. For the ARC subset considered, one needs to retain at least 500 facts to see a noticeable improvement in QA accuracy.

• The question-driven Mt extraction method can provide performance gains on par with a clean, hand-annotated corpus (WorldTree)

• When extracting core facts from $\mathcal{F}$, the choice of optimization method doesn't have a noticeable effect on QA, though each method outperforms random selection.

## 5.3 RELEVANCE

Finally, to complement our empirical investigation, we perform two subjective assessments of the contents of the microtheories, the first using (expert) human experts and the second using an LLM-as-judge. This provides an orthogonal evaluation of whether our microtheory-building process is indeed extracting the more important, central facts about the topic. We are particularly interested in whether, and by how much, our microtheory condensation and optimization methods (§3.2.3) are selecting important facts from the general fact pool $\mathcal{F}$.

---

[7]This decrease results from the irrelevant facts distracting the entailment search, hurting the grounding rate.

### 5.3.1 EXPERT RELEVANCE ANNOTATIONS

First, we collect human judgments of the relevance of Mt facts. We use the MedQA medical domain due to easy access to domain experts. We asked experts how useful they would find the Mt facts when studying for and taking USMLE tests. We recruited two senior medical students to annotate the general relevance of facts from different MedQA microtheories.

We worked with the annotators to develop a 5-point rubric for scoring how essential a fact is to the core content that appears on the USMLE. Annotation details and the full rubric are in §J.

Average relevance ratings are shown in Figure 8. Both optimization approaches outperform a random baseline by at least half a grade point; $Mt_{PC}$ is 0.6-1.2 higher, and never drops below a score of 4. (The annotation rubric score defines a score of 4 as containing relevant exam content, though either too general or unspecific to sufficiently answer an exam question.) In contrast, a randomly selected fact from the full pool frequently scores below 3.5, often containing context or background information less directly useful for answering exam questions. This suggests that the Mt extraction methods successfully distill out the most topically relevant facts from the general knowledge pool, successfully selecting the information most likely to support exam performance.

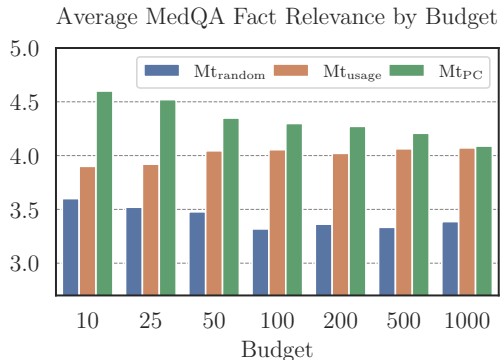

Figure 8: Average expert-annotated task relevance score for microtheory facts. *PC* facts are more relevant than *random* and *usage* until the 1000-fact budget.

### 5.3.2 AUTOMATIC PER-QUESTION RELEVANCE

As well as evaluating the Mt as a whole, we also explore whether Mts are relevant on a *per-question* basis, to see how often the Mt has *something* useful to contribute when answering a new question. We ran a study to assess this, using an LLM as a judge. Although necessarily approximate using an LLM as the scorer, it provides further insights into our Mts. First, we say a Mt is *relevant* if it contains *at least one* fact that is relevant for getting the question right, i.e., differentiating the correct answer from any incorrect options. To determine if a fact is relevant, we use an existing scoring rubric developed by Jansen et al. (2021) for rating a fact's relevance to a question on a 0-5 scale, and here ask `gpt-4o-2024-08-06` to make that judgment (see §K for the rubric and the full prompt). We then measure the rate at which at least one 5-rated fact is in the Mt.[8] This provides a soft indicator that the Mt contains information that would be core to a basis for the correct answer.

Figure 9 shows the change in per-question relevance rate as we increase $n$ for $Mt_{PC}$. Our main observations are that **$Mt_{PC}$ is substantially more relevant than $Mt_{random}$** (see Figure 23 for other methods) and that the **ARC Mts contain 5/5 relevant facts at a much higher rate than MedQA.** For example, the 100-$Mt_{PC}$ obtains a 72.3% relevance rate for ARC, while for MedQA, this rate drops to 30.1%. Similarly, the 1000-$Mt_{PC}$ has around 90% relevance for ARC but only 70% for MedQA. Even looking at the entire fact pool $F$, it is only 93% relevant for MedQA (right-hand bar in Figure 9), while nearly 100% relevant for ARC. This suggests that it is easier to find highly reusable, general principles in ARC's domain (science), e.g., basic laws of electricity, than in MedQA (medicine). Discussions with our medical experts reinforce this: while there are also general principles in medicine, e.g., "The body strives to maintain a stable internal environment," these are fewer and far between, and the USMLE questions instead often probe a student's long-tail knowledge of highly specific medical facts (e.g., symptoms of Hippel-Lindau disease). This suggests that some domains may be more amenable to distilling into microtheories than others, and explains the more limited gains seen earlier with MedQA compared with ARC.

## 6 HOW COMPLETE ARE MICROTHEORIES?

How many training questions do we need to construct a "complete" microtheory? This is an important, practical concern for anyone wanting to extract microtheories for a new domain. In one sense, a microtheory is never complete, as there will always be topical questions that touch areas uncovered in training. However, we can make progress on this question in a different way. First, given a

---

[8]For practical reasons, we only assess the top 270 facts retrieved from the Mt by the entailment engine (for Mts larger than 270). The engine uses an SBERT retrieval encoder fine-tuned on science QA support facts.

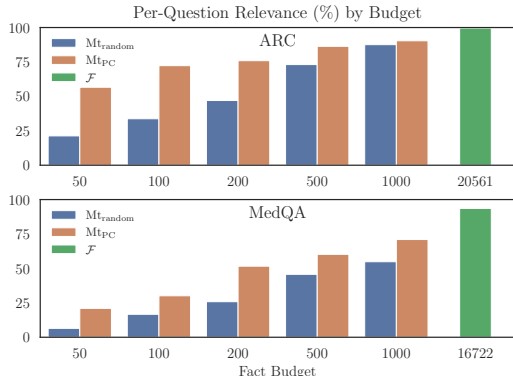

Figure 9: Per-question relevance rate for $n$-$MT_{PCS}$ for varying $n$s. The rate is much higher in the ARC domain than MedQA; at 1000 facts, the Mt is *at all* relevant to only 2/3 of MedQA questions but 90% of ARC.

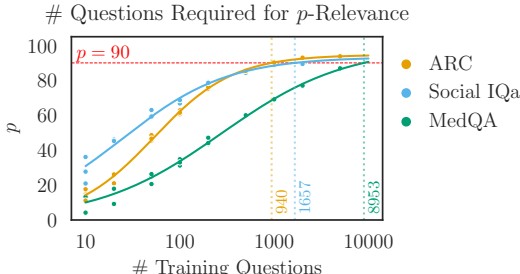

Figure 10: Number of training questions (x-axis) needed to create an $\mathcal{F}$ that is relevant to test questions $p$% of the time (y-axis). While datasets like ARC and Social IQa (Sap et al., 2019) need 900-1600 questions for 90-relevance, but MedQA needs far more (8900).

distribution over topical questions, we say that a microtheory is *relevant* to a question if it contains at least one fact that is core to the (correct) answer according to the method described in §5.3.2. We then say a microtheory is partially ($p$ = e.g., 90%) complete if it is relevant to an arbitrary sampled question with probability $p$. Finally, we can predict this *p-relevance* by generating and extrapolating from microtheory learning curves such as in Figure 10.[9] For example, to obtain an Mt with a $p$-relevance of 90%, we will likely need 940 training questions for ARC, but 8953 training questions for MedQA.[10] Conversely, given a training set of size $n$, we can predict the likely $p$-relevance of the Mt from the curve. Thus we would expect 100 questions to provide enough knowledge to be ($p$=60)-relevant for ARC but only 30-relevant for MedQA.

$p$-relevance gives practitioners a way of predicting whether Mts will be helpful for their new dataset: determining how much data they would need to collect for a new domain for the Mt to be frequently useful for the topic. If this number is too high, then Mts might not be a viable solution for the domain – this is likely the case for many datasets for which there is less of a shared body of knowledge that is reapplied to many questions, e.g. HotPotQA (Yang et al., 2018) and Natural Questions (Kwiatkowski et al., 2019), where each question asks about a unique Wikipedia factoid. The number of training questions also has a large effect on the amount of compute used to construct a microtheory, since calling the entailment engine to collect usage statistics and distill the most generalizable facts requires many LLM calls per question. The smaller the training set necessary to obtain high $p$ values, the less LLM calls; running the engine on 8953 MedQA questions to obtain 90-relevance would be making 100,000s of LLM calls. Future work might explore which kinds of QA datasets are realistically conducive to Mts.

## 7 CONCLUSION

While LLMs have strong reasoning abilities, it is often unclear how their beliefs recurrently interact to answer questions on a topic, a significant barrier for understanding and deploying models. To address this, we have presented a method for materializing a model's latent, topical knowledge into a *microtheory*, a linguistic analog of logical microtheories, articulating the model's core, reusable knowledge about the topic. In contrast to explanation methods that show snippets of knowledge supporting individual answers, ours is the first method that attempts to distill a model's *overall understanding* of a topic. Our evaluations suggest that, when added to a general corpus, microtheories can significantly improve a model's ability to ground its answers, i.e., show how they are systematically entailed by corpus documents, as well as improving the accuracy of those grounded answers. In addition, human and LLM-as-judge assessments of the microtheories suggest they contain a high proportion of critical, topical facts, and that our construction process successfully identifies many of those important facts from a general pool. Together, these suggest that microtheries are an efficient distillation of model's topic-relevant knowledge, and can provide both performance gains and, for the first time, an interpretable window into a model's topical knowledge.

---

[9]We fit ($R^2 = 1.00$) a modified Hill equation with an additional scaling parameter: $y = \frac{V_{\max} \cdot x^n}{K^n + x^n} \cdot s$.

[10]Unlike in Figure 9, we use the original train/test sets here. We compute the per-question relevance of $\mathcal{F}$ prior to any reductions, thus posing an upper bound on the relevance of any Mt distilled from the larger pool.

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

# A  USAGE OPTIMIZATION DETAILS

## A.1  $n$-MT$_{\text{QC}}$ LINEAR PROGRAM

We set the objective function to achieve the following goals, ordered by precedence:

- Maximize the coverage of questions by at least one proof tree.

- Maximize the number of proof trees covered by the selected facts.

- Minimize the number of facts included in the new theory.

We define the objective as

$$\min \left( \sum_{i=1}^{|\mathcal{C}|} x_i - \sum_{j=1}^{d} \left( \omega z_i + \sum_{k=1}^{|\mathcal{L}_i|} y_{kj} \right) \right) \tag{5}$$

Where $x_i$ is a binary variable indicating whether fact $i$ is included in the Mt, $y_{kj}$ is a binary variable indicating whether basis $L_{kj}$ for question $j$ is covered by the selected facts, and $z_j$ is a binary variable indicator whether question $q_j$ has at least one basis fully covered. We set $\omega$ to be sufficiently high to dwarf the other terms, using them only for breaking ties.

We impose the following constraints:

- At most $n$ facts are kept.

$$\sum_{i=1}^{|\mathcal{C}|} x_i \leq n \tag{6}$$

- Each $L_{kj}$ is "covered" iff all its facts are kept.

$$\forall L_{kj}, \quad \sum_{i=1}^{|L_{kj}|} x_i \geq y_j \cdot |L_{kj}| \tag{7}$$

- Each question is "covered" if at least one of its bases is "covered"

$$\forall \mathcal{L}_j, \quad \sum_{j=1}^{|\mathcal{L}_j|} y_j \geq z_k \tag{8}$$

## A.2  $n$-MT$_{\text{PC}}$ ILP IMPLEMENTATION

We set the objective function to achieve the following goals, ordered by precedence:

- Maximize the partial coverage of each question by the selected facts.

- Minimize the number of facts included in the new theory.

We define the objective as

$$\min \left( \sum_{i=1}^{|\mathcal{C}|} x_i - \sum_{j=1}^{d} \omega p_j \right) \tag{9}$$

Where $x_i$ is a binary variable indicating whether fact $i$ is included in the Mt, $p_j$ is a continuous variable indicating the maximal partial coverage of question $j$ by the selected facts, and $y_{kj}$ is a continuous variable indicating the coverage of basis $L_{kj}$ for question $j$ by the selected facts. $z_{kj}$ is a binary indicator of whether a given $L_{kj}$ is the highest coverage of question $j$ attained by the solution.

We impose the following constraints:

- At most $n$ facts are kept using Equation 6.

- Each $L_{kj}$ is "covered" by the proportion of its facts that are kept.

$$\forall L_{kj}, \quad y_{kj} = \frac{1}{|L_{kj}|} \sum_{i \in L_{kj}} x_i \tag{10}$$

- Two constraints that enforce $p_j$ equals the highest parital coverage for question $j$ using the Big M method: The maximal partial coverage of each question is at least as large as the coverage of each of its bases.

$$\forall \mathcal{L}_j, \quad p_j \geq y_{kj} - M(1 - z_{kj}) \quad \forall k \tag{11}$$

and the maximal partial coverage of each question does not exceed the coverage of each of its bases.

$$\forall \mathcal{L}_j, \quad p_j \leq y_{kj} + M z_{kj} \quad \forall k \tag{12}$$

- Exactly one $z_{kj}$ is active per question.

$$\forall \mathcal{L}_j, \quad \sum_{k=1}^{|\mathcal{L}_j|} z_{kj} = 1 \tag{13}$$

## B  QUESTION/TOPIC CLUSTERING ALGORITHM

To create our topic-specific mini-splits of ARC and MedQA, we use a hierarchical clustering algorithm that gathers questions targeting similar knowledge by iteratively prompting an LLM to 'caption' questions with core targeted knowledge sentences and then clustering them using SBERT. Algorithm 1 depicts our implementation of the question clustering algorithm used to extract subsets of a broader QA dataset that target similar underlying core knowledge. For the embedding clustering algorithm we use the SentenceTransformer-based community detection algorithm that clusters sentences greedily according to some minimum cosine similarity threshold, which we set to $0.6$.

---

**Algorithm 1** Two-Step Clustering Algorithm for Question Grouping by Core Topic

---

1: **procedure** TWOSTEPCLUSTERING($Q$) a Hierarchically clustered questions with topic and hypertopic labels:
2:      $core\_facts$ = []
3:      **forall** $q_i \in Q$ **do**
4:          $core\_fact$ = LLM.EXTRACTCOREFACT($q_i$)
5:          $core\_facts$ += $core\_fact$
6:      **end for**
7:      $clusters$ = SBERT.CLUSTER($core\_facts$)
8:      $topic\_labels$ = []
9:      **forall** $c_j \in clusters$ **do**
10:          $topic\_label$ = LLM.GENERATETOPICLABEL($c_j$)
11:          $topic\_labels$ += $topic\_label$
12:      **end for**
13:      $topic\_clusters$ = SBERT.CLUSTER($topic\_labels$)
14:      $hypertopic\_labels$ = []
15:      **forall** $tc_k \in topic\_clusters$ **do**
16:          $hypertopic\_label$ = LLM.GENERATEHYPERTOPICLABEL($tc_k$)
17:          $hypertopic\_labels$ += $hypertopic\_label$
18:      **end for**
19:      **Assign** $topic\_labels$ **and** $hypertopic\_labels$ **to corresponding questions in** $Q$
20:      **return** $Q$ with hierarchical topic and hypertopic labels

---

## C  FACT EXTRACTION PROMPTING DETAILS

The LLM prompt to generate relevant facts for a given $Q_i$ is a few-shot learning prompt using exemplars dynamically retrieved[11] from EntailmentBank. Each exemplar's fact list comprises the gold WorldTree (Xie et al., 2020) facts from the unstructured explanation graph, and its entailment tree is the gold entailment tree that uses a subset of the facts. Thus, while $Q_i$ might not be in the same ScienceQA domain as the exemplars, we cue the LM to generate *WorldTree-like* facts that should serve similar roles as argumentative bases in entailment trees. We discard the model-generated entailment tree and take the generated fact list as candidates to be added to the fact pool.

Figure 11 shows an example dynamically constructed in-context learning prompt for fact extraction. Figure 12 shows an example output from Mixtral-8x22b-Instruct-V0.1 for a MedQA question.

---

[11]We use BM25 (Robertson et al., 1995) for retrieval using the question text as the query.

```
You are an expert theory generation system that lists facts about a problem area that will help solve a
    reasoning question. Given a question, you generate QUERY statements for each of the possible answers to
    the question. The queries should be complete statements that exactly match the answer choices as closely
    as possible.

Then, you generate a theory comprising simple FACTS about the world. These should be statements that you
    believe to be logical and true about the world that will help you prove the queries. A FACT is either a
    generic statement about the world ("birds can fly") or a specific statement about the context that helps
    support the proof of the query ("Tweety is a bird").

For a QUERY, you generate at least 6 statements.

Once you generate the theory, you show how one of the QUERYs logically follows from the context. The selected
    QUERY corresponds to the answer choice that you believe is correct. You can only choose one QUERY as
    correct.

To show your reasoning, you generate a PROOF of your chosen QUERY hypothesis. Your PROOF is an "entailment
    tree" that composes the FACTS of your theory in order to show how the query is compositionally entailed
    by them. Each step in the tree is a composition of two or more premises into a new conclusion, which
    should be used as a premise in future steps. The last step compositionally proves the hypothesis. When
    you reuse the new conclusion in a later step, you refer to it by its id. Each step is separated by a
    newline.

###
QUESTION: Which of the following actions is most likely part of a test to find the hardness of a mineral
    sample? (A) heating the sample on a hot plate (B) scratching the sample with a nail (C) hitting the
    sample with a hammer (D) shining a bright light on the sample

QUERIES:
(QUERY A) Heating the sample on a hot plate is most likely part of a test to find the hardness of a mineral
    sample.
(QUERY B) Scratching the sample with a nail is most likely part of a test to find the hardness of a mineral
    sample.
(QUERY C) Hitting the sample with a hammer is most likely part of a test to find the hardness of a mineral
    sample.
(QUERY D) Shining a bright light on the sample is most likely part of a test to find the hardness of a mineral
    sample.

THEORY:
(FACT 1) a mineral is a kind of material
(FACT 2) hardness is a measure of a mineral 's ability to resist scratching
(FACT 3) an example of breaking something down is scratching something
(FACT 4) a material that is soft can be broken down by a material that is hard
(FACT 5) testing often requires measuring
(FACT 6) a nail is a kind of object
(FACT 7) hardness is a property of a material / an object and includes ordered values of malleable / rigid
(FACT 8) hardness can be used to identify minerals
(FACT 9) a mineral is a kind of solid / natural material
(FACT 10) nails are often made of iron
(FACT 11) scraping an object may cause small particles to break off of that object
(FACT 12) if a mineral can be scratched by a fingernail then that mineral is soft
(FACT 13) hardness is a kind of physical property
(FACT 14) hardness is an intensive property
(FACT 15) soft is a kind of hardness
(FACT 16) experimentation requires measuring
(FACT 17) an event is a kind of action
(FACT 18) if a material scratches easily then that material has low hardness
(FACT 19) a mineral is a kind of object
(FACT 20) mineral nutrient is a kind of nutrient
(FACT 21) objects are made of materials / substances / matter
(FACT 22) rock is made of minerals
(FACT 23) a process is made of a series of actions
(FACT 24) if an object is made of a material then that object has the properties of that material
(FACT 25) testing / identifying the streak of a mineral requires the powdered form of that mineral

PROOF of Scratching the sample with a nail is most likely part of a test to find the hardness of a mineral
    sample.:
(STEP 1) a material that is soft can be broken down by a material that is hard & an example of breaking
    something down is scratching something -> int1: a material that is soft can be scratched by a material
    that is hard
(STEP 2) int1 & a mineral is a kind of material -> int2: if one mineral can scratch another mineral then that
    other mineral is softer than that one mineral
(STEP 3) hardness is a measure of a mineral 's ability to resist scratching & int2 -> hypothesis: Scratching
    the sample with a nail is most likely part of a test to find the hardness of a mineral sample.

ANSWER: B

###
QUESTION: {question}

QUERIES:
{queries}

THEORY:
```

Figure 11: TheoryCoT in-context learning prompt for extracting WorldTree-like facts from an LLM for a given question.

```
###
QUESTION: A 47-year-old woman with chronic epigastric pain comes to the physician because of a 1-month history
       of intermittent, loose, foul-smelling stools. She has also had a 6-kg (13-lb) weight loss. She has
       consumed 9-10 alcoholic beverages daily for the past 25 years. Seven years ago, she traveled to Mexico
       on vacation; she has not been outside the large metropolitan area in which she resides since then. She
       appears malnourished. The stool is pale and loose; fecal fat content is elevated. An immunoglobulin A
       serum anti-tissue transglutaminase antibody assay is negative. Further evaluation is most likely to show
        which of the following? (A) Trophozoites on stool microscopy, (B) Pancreatic calcifications, (C)
       Villous atrophy of duodenal mucosa, (D) Positive lactulose breath test

QUERIES:
(QUERY A) Further evaluation is most likely to show trophozoites on stool microscopy.
(QUERY B) Further evaluation is most likely to show pancreatic calcifications.
(QUERY C) Further evaluation is most likely to show villous atrophy of duodenal mucosa.
(QUERY D) Further evaluation is most likely to show a positive lactulose breath test.

THEORY:
(FACT 1) Chronic epigastric pain can be a symptom of various gastrointestinal disorders.
(FACT 2) Intermittent, loose, foul-smelling stools can be a sign of malabsorption.
(FACT 3) Significant weight loss can be a sign of malnutrition or malabsorption.
(FACT 4) Long-term, heavy alcohol consumption can lead to pancreatitis and malabsorption.
(FACT 5) Travel to certain regions can expose individuals to parasites that cause gastrointestinal issues.
(FACT 6) A negative immunoglobulin A serum anti-tissue transglutaminase antibody assay rules out celiac
       disease.
(FACT 7) Malabsorption can lead to elevated fecal fat content.
(FACT 8) Pale and loose stools can be a sign of malabsorption.
(FACT 9) Pancreatic calcifications can be a sign of chronic pancreatitis.
(FACT 10) Villous atrophy of duodenal mucosa can be a sign of celiac disease or other gastrointestinal
       disorders.
(FACT 11) A positive lactulose breath test can indicate small intestinal bacterial overgrowth (SIBO).

PROOF of Further evaluation is most likely to show pancreatic calcifications.:
(STEP 1) Long-term, heavy alcohol consumption can lead to pancreatitis and malabsorption & Significant weight
       loss can be a sign of malnutrition or malabsorption -> int1: The patient's long-term, heavy alcohol
       consumption may have led to pancreatitis and malabsorption, causing significant weight loss.
(STEP 2) int1 & Intermittent, loose, foul-smelling stools can be a sign of malabsorption -> int2: The patient'
       s intermittent, loose, foul-smelling stools are likely due to malabsorption caused by pancreatitis.
(STEP 3) int2 & Pancreatic calcifications can be a sign of chronic pancreatitis -> hypothesis: Further
       evaluation is most likely to show pancreatic calcifications.
```

Figure 12: Example TheoryCoT output for a MedQA question. The generated proof steps are thrown out and the generic facts are added to the fact pool $\mathcal{F}$. Context-specific facts are discarded.

## D    ENTAILMENT ENGINE IMPLEMENTATION DETAILS

To collect training proofs, we explore up to depth 2 for ARC hypotheses (these are easier to prove) and up to 40 expansions for MedQA.

### D.1    RETAINING INFERENCES

When collecting the bases $L_{ij}$, we want the entailment engine to best identify which facts are most commonly reused across questions. Since the engine has the potential to miss certain combinations, we add a mechanism to encourage reusing inferences from earlier training questions $Q_1 \ldots Q_{i-1}$ when finding bases for $Q_i$. After each call to ENGINE($h_i, \mathcal{C}; Q_i$), we take all intermediate sub-hypotheses grounded by the engine to facts in $\mathcal{C}$ in the process of proving $h_i$, some $\mathcal{S}_i$, and add them to the fact pool for all future questions $j > i$. Whenever the engine uses any subhypothesis $s \in S_i$, we replace it with all the ways the engine grounded $s$ using bases in $\mathcal{C}$. E.g., if the engine returns basis $[f_1, s]$ and $s$ was itself previously grounded by the sets $[f_2, f_3], [f_2, f_4]$, we replace $[f_1, s]$ with $[f_1, f_2, f_3]$ and $[f_1, f_2, f_4]$.[12]

## E    QUALITATIVE EXAMPLES AND ANALYSIS

Figure 13 and Figure 14 list the top-10 most frequent facts used for training questions, as well as a sample of 10 from the long tail of facts used for only one question. We also show which of these facts are retained in the partial coverage 100-Mt$_{PC}$. The top ARC facts are core pieces of knowledge very similar to those that appear in the hand-crafted WorldTree corpus; many of the facts in MedQA are about chronic obstructive pulmonary disease (COPD), reflecting that the Smoking topic is the largest in the training set and that COPD is a common medical condition associated with smoking.

Figure 15 depicts a subgraph of a much larger network of entailments connecting statements in $\mathcal{C}$ to 18 of the training hypotheses. It shows which of these statements were retained by the partial coverage Mt for their support of multiple related hypotheses. The subgraph illustrates 4 cliques; one for hypotheses about the gravitational force between opbjects, and 3 about the force, acceleration and mass of objects in motion. The core statements retained in the Mt are mainly versions of Newton's second law of physics, $F = m \cdot a$ and Newton's Law of universal gravitation. While a singular fact

---
[12]To limit combinatorial explosions, we produce a maximum of 1000 bases per question via this "unfolding".

| Fact | # Qs | Mt$_{PC}$ |
|---|---|---|
| any feature which is determined by genes that can be transmitted from parent to offspring during reproduction is considered an inherited trait | 23 | Yes |
| the earth turns on its axis from west to east once every 24 hours | 22 | Yes |
| sedimentary rocks are most often formed by the compaction and cementation of sediment, which is material transported and deposited by wind, water, or ice | 20 | Yes |
| the tilt of earth's axis is the reason for the changing amount of sunlight that each hemisphere receives, which in turn is the driver for seasons | 20 | Yes |
| genes are the instructions for physical traits that pass from parents to offspring | 20 | Yes |
| Though Earth's revolution around the Sun does affect the duration of daylight and darkness, the basic existence of the two phenomena is due to Earth's rotation on its axis | 19 | Yes |
| inheriting means receiving genetic information and traits from a parent or parents | 19 | No |
| Physical traits of an organism, such as fur color, eye color, and size, can be genetically passed from parents to offspring | 19 | Yes |
| The rotation of the Earth regulates the daily cycle of day and night, not the seasonal cycle | 19 | Yes |
| the rotation of the Earth on its axis is the movement of the Earth around a central point, once every 24 hours | 18 | No |
| Whales utilize ocean currents and temperature gradients to guide their migration routes | 1 | No |
| Mutations can be caused by various factors such as exposure to radiation, certain chemicals, or errors during DNA replication | 1 | No |
| a tree requires sunlight to grow | 1 | No |
| lava beds are igneous rock, formed from cooled lava | 1 | No |
| wilting is a survival mechanism, limiting water loss by reducing the surface area exposed to the sun | 1 | No |
| Mendel's experiments with pea plants were focused on heredity | 1 | No |

Figure 13: Top-10 most used and random 10 least used (at least once) facts in the ARC microtheory. The rightmost column depicts whether the fact is retained in the "partial coverage" microtheory (Mt$_{PC}$) that reduces redundancy while maximizing training coverage.

| Fact | # Questions | Mt$_{PC}$ |
|---|---|---|
| The severe smoker with an estimated 40 pack-year history has a high risk for chronic obstructive pulmonary disease (COPD) and lung cancer | 45 | Yes |
| The man's history of long-term smoking puts him at increased risk for cardiovascular and lung diseases, including pulmonary embolism | 34 | Yes |
| a heavy smoker with a 25-pack-years history is at a high risk for lung diseases | 32 | No |
| Chronic obstructive pulmonary disease (COPD) is generally seen with a history of smoking and presents with symptoms such as shortness of breath, progressive dyspnea, cough, and sputum production | 31 | Yes |
| a 50 pack-year smoking history contributes to a multitude of health risks, including lung cancer | 27 | No |
| findings such as blood and protein in the urine, as well as high serum urea nitrogen and creatinine levels, indicate kidney damage | 25 | Yes |
| A 40 pack-year smoking history, hypertension and diabetes make one more likely to have COPD | 25 | No |
| a man with a 50 pack-year smoking history has a very high risk of developing health issues related to smoking | 25 | No |
| long-term exposure to lung irritants, particularly tobacco smoke, is the primary risk factor for developing chronic obstructive pulmonary disease | 24 | No |
| A patient's history of smoking cigarettes daily for 25 years can have a significant impact on their respiratory function, leading to problems such as shortness of breath, especially after a surgical procedure | 22 | No |
| A rapidly enlarging scalp lesion in a child, especially if blanching with pressure, is suggestive of a vascular lesion such as a strawberry hemangioma | 1 | No |
| losing a job due to issues such as tardiness is likely due to some negative circumstance impacting the individual's life, such as alcohol abuse | 1 | No |
| Bloody diarrhea can be a symptom of amoebic dysentery caused by amoebic infection | 1 | No |
| fever is a common symptom in vasculitis because of the inflammation | 1 | No |
| Normal-pressure hydrocephalus is characterized by cognitive changes, gait problems, and urinary incontinence | 1 | No |
| Blood pressure of 197/124 mm Hg is considered dangerously high and needs to be lowered immediately | 1 | No |
| Creatinine phosphokinase is an enzyme found in the heart, brain, skeletal muscle and other tissues, its levels increase when muscle or heart cells are injured | 1 | No |
| the liver breaks down bilirubin so it can be removed from the body in the stool | 1 | No |
| Smoking and unprotected sex increase the risk of various infections, including urinary tract infections | 1 | No |
| Regular kneeling can put excessive pressure on prepatellar bursa, potentially causing inflammation and pain | 1 | No |

Figure 14: Top-10 most used and random 10 least used (at least once) facts in the MedQA microtheory. The rightmost column depicts whether the fact is retained in the "partial coverage" microtheory (Mt$_{PC}$) that reduces redundancy while maximizing training coverage.

describing the law might theoretically serve the same purpose in proving all these hypotheses, We

observe that the entailment engine prefers to use separate facts representing different interpretations of the law (e.g. how to calculate net force, vs how to calculate the force required to move an object). This may also be a result of imposing a depth-2 limit of the training proof search. It would perhaps require more "legwork" by the entailment engine to construct a chain of more than 2 entailments from a single statement, e.g. just 'force equals mass times acceleration,' to each hypothesis.

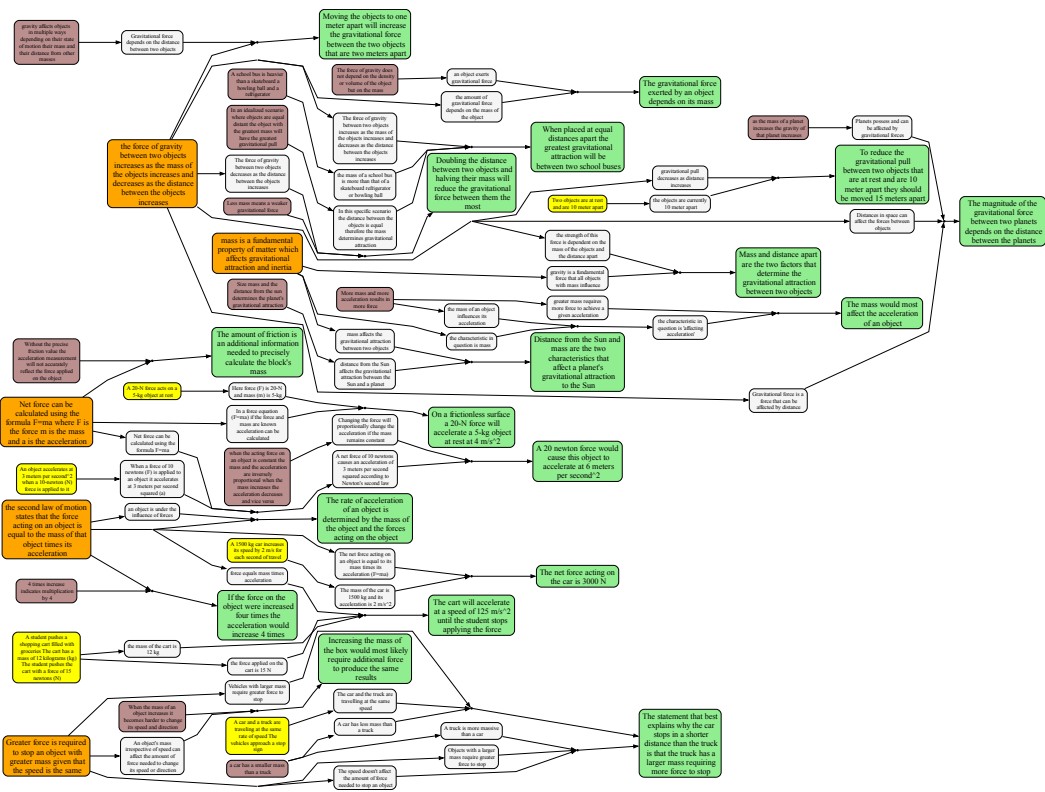

Figure 15: Subgraph showing some of the statements from the 200-Mt$_{PC}$ ARC microtheory ( orange ) and their influence on proving training hypotheses ( green ) in different question contexts ( yellow ). The partial coverage (PC) algorithm selects facts from $\mathcal{C}$ that provide coverage for multiple questions. Facts from $\mathcal{C}$ that are specific to only one or two questions ( maroon ) are not selected. Each dot (·) represents an entailment; inferences ( grey and green ) can be supported by alternative entailments in the form of multiple entering edges. In this instance, the PC algorithm shows to retain 5 facts mostly related to Newton's second law, which is core to multiple subtopics in grade school physics tests (motion, friction, gravitational attraction). Only the orange nodes are retained in the microtheory for test time inference; all other statements are discarded.

Figure 16 shows a similar subgraph for the MedQA microtheory, this time illustrating the facts retained by the full question coverage 100-Mt$_{QC}$. Hypotheses are fully thus grounded to Mt statements and the context passages of their respective questions. This graph also illustrates the effect of retaining inferences during the serial computation of the $L_{kj}$ leaf sets.

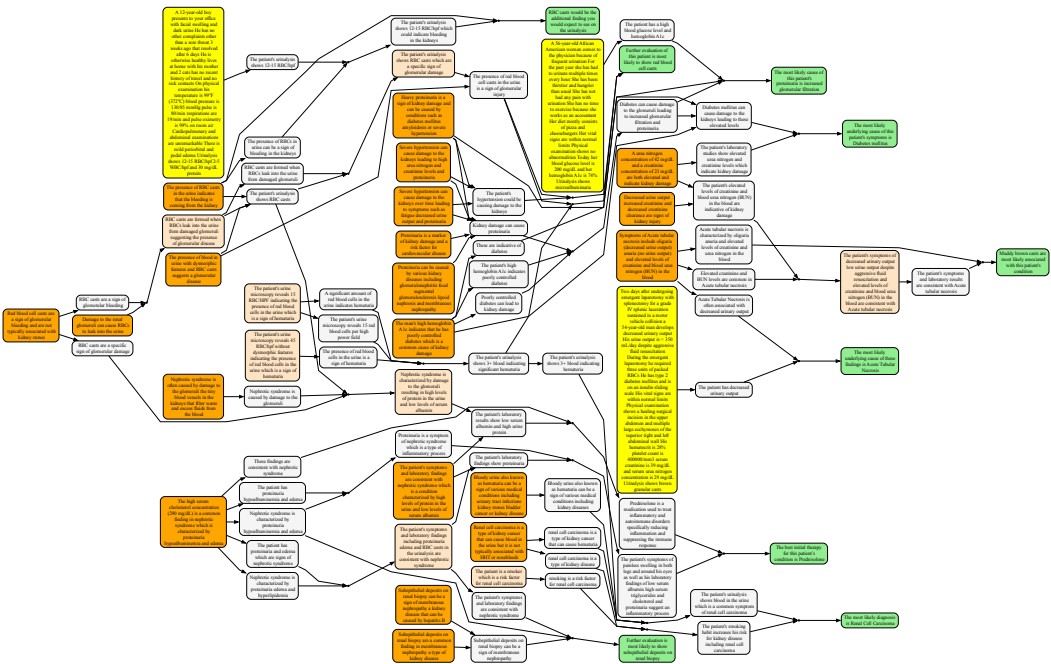

Figure 16: Subgraph showing some of the statements from the 100-Mt$_{QC}$ MedQA microtheory ( orange ) and their influence on proving training hypotheses ( green ) in different question contexts ( yellow ) related to clinical cases. Unlike the PC approach, the question coverage (QC) approach ensures full grounding of each training hypothesis using facts from the microthery. As MedQA hypotheses often require many entailment hops, the inference retention mechanism described in §D.1 helps prove hypotheses using inferences ( light orange ) grounded from previous questions, some of which are omitted for space.

## F    RETRIEVAL IMPLEMENTATION DETAILS

The Wikipedia corpus used in §4 is structured as a flat index (i.e., a very long list) of paragraph chunks– specifically, the 2021-01- 20 version of Wikipedia with 100 word chunks. For similarity, we split the textbooks into 100 word chunks as well. We retrieve paragraphs using two-phase retrieval; we use BM25 (Robertson et al., 1995) for first stage retrieval and rerank using Sentence-Transformer's ms-marco-MiniLM-L-12-v2 (Reimers & Gurevych, 2019).

When we retrieve facts from both the Microtheory and Wikipedia (or Wikipedia+Textbooks), we append the sets in that order. When we retrieve from Wikipedia+Textbooks, we interleave the results one-by-one to give equal precedence to the two sources. These design choices were chosen based on empirical performance.

## G    MIXTRAL RESULTS ON MEDQA

To demonstrate our work is not closed-model dependent, this section replicates the MedQA experiments from the paper using Mixtral-8x22B-Instruct-v0.1 (Jiang et al., 2024) as the source of the microtheories instead of GPT-4. Figure 17, Figure 18, Figure 19, and Figure 20 each show Mixtral results analogous to an experimental result from the main paper using GPT-4. We have commented on the similarities and differences with the original result in the caption of each figure.

## H    ADDITIONAL FACT USAGE METRICS

In addition to the grounding rate (§5.1), human-annotated relevance rate (§5.3.1), and automated relevance rate (§5.3.2), this section shows a pair of additional metrics for our considered microtheories: (1) the fraction of all facts in a microtheory used by the entailment engine for at least one

| Mixtral-8x22b Results | MedQA |
|---|---|
| **Dataset Details** | |
| # Questions | 641/94/186 |
| # Topic Clusters | 4 |
| Ext. Corpus | Wiki+Textbooks |
| **Training Extraction Results** | |
| $\|\mathcal{F}\|$ | 11,082 |
| $\|\mathcal{C}\|$ | 8,183 |
| # Qs with Proof | 476 (74%) |
| Min #Fs to Cover | 918 |
| Fact/Q Ratio | 1.91 |

Figure 17: Results of microtheory extraction from training data using Mixtral-8x22b as the underlying LLM. The last two rows result from the "min-fact" LP. Compared with GPT-4, Mixtral generates 5000 fewer unique facts (11K vs 16K) and finds proofs for 14% fewer questions.

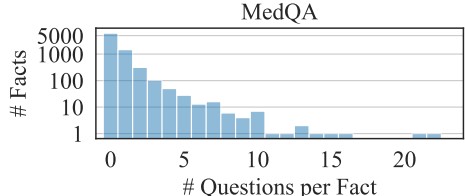

Figure 18: Histogram of training question per fact in $\mathcal{C}$ and 650 MedQA training questions using Mixtral-8x22b as the underlying LLM. While multiple GPT-4 generated facts are used for 25 or more training questions, no Mixtral-generated fact is used more than 22 times.

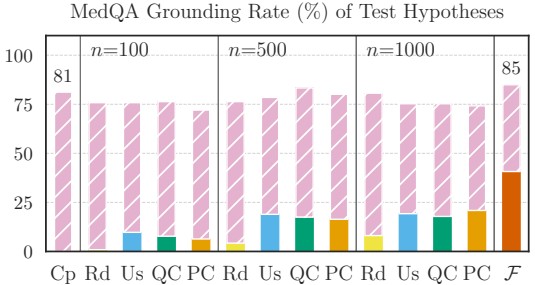

Figure 19: % of test hypotheses fully groundable after adding various Mixtral-generated microtheories ($n$-sized **R**and**o**m, **Us**age, **QC**, & **PC** Mts) to a base **C**or**p**us. We show the fraction of leaves grounded in the Mt (solid) and the base corpus (striped).

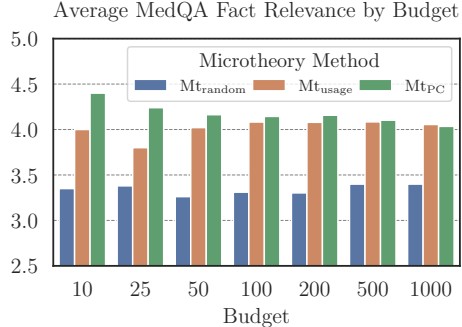

Figure 20: Average expert-annotated task relevance score for Mixtral-generated microtheory facts.

test question, and (2) the average number of test questions for which each fact in the microtheory is used.

Figure 21 shows the fraction of microtheories used at least once at test time, and Figure 22 shows the average number of questions for which any single Mt fact is used. In both cases, the $Mt_{usage}$, $Mt_{QC}$, and $Mt_{PC}$ have higher usage than the random sample baseline, while the full fact pool $\mathcal{F}$ scores very low, since most of the facts in the full pool are never used at test time. Expectedly, the $Mt_{usage}$ approach scores highest on both metrics, given this is what it was optimized for; we sorted facts by their usage counts at training time (the same metric measured in Figure 22) and took the top-$n$ as the Mt. We note that this does not directly correlate with Mt quality since, as we discussed in §3.2.3, sorting only by per-fact usage fails to remove functional redundancies. The test grounding rates in §5.1 should be considered a more encompassing metric.

# I  EXTENDED RELEVANCE RATE RESULTS

Figure 23 shows the rate at which the considered microtheory methods contain 5-rated relevant facts for the test sets in the topic mini-splits of ARC and MedQA.

# J  EXPERT RELEVANCE ANNOTATION DETAILS

We sampled 25 facts (fewer when necessary) from Mts with 70 budgets of size ranging from 10 to 1000. We evaluated the $Mt_{usage}$, $Mt_{PC}$, and $Mt_{random}$ approaches. Both annotators rated each fact independently and then reconciled the (rare) disagreements. We computed agreement metrics across the full set of 314[13] annotated facts, finding a Krippendorff $\alpha$ of .81, Cohen's $\kappa$ of .809, and raw agreement rate of .863.

---

[13]This number is not $7 \times 3 \times 25$ because some Mts were smaller than 25 facts, the $n$-$Mt_{usage}$s are all supersets of each other, and many other samples contained high overlap due to the extraction algorithm behavior.

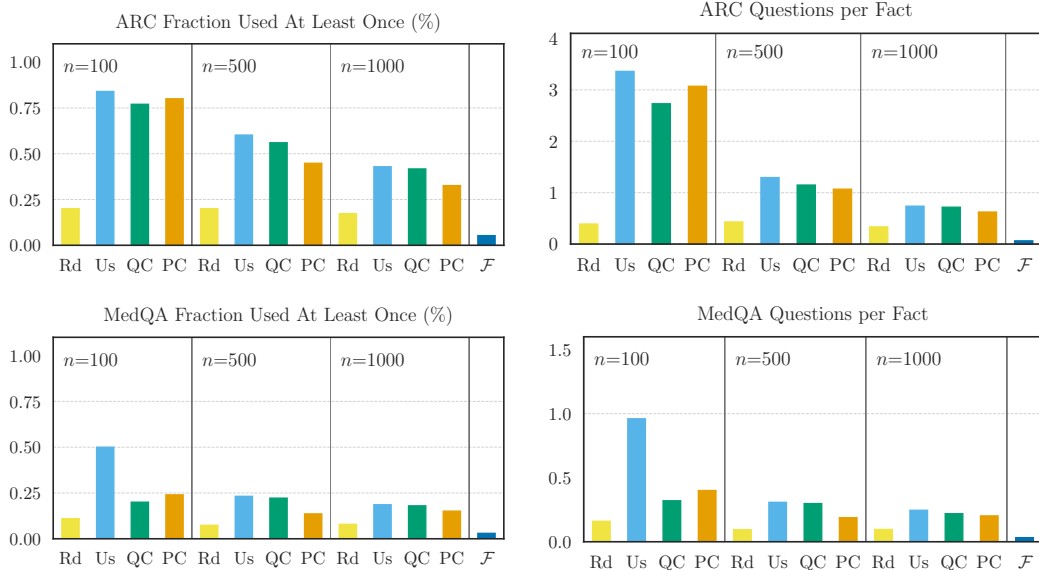

Figure 21: Fraction of a microtheory used in an entailment proof for at least one test question (out of 249 for ARC, 186 for MedQA). All distillation methods (n-sized **Us**age, **QC**, & **PC** Mts) are used substantially more frequently than the random sample baseline, with the **Us**age having the highest fraction. The fraction used for the full raw fact pool $\mathcal{F}$ is very low.

Figure 22: Average number of test questions (out of 249 for ARC, 186 for MedQA) per fact in a microtheory. Distilled microtheories are used in more questions than the random baseline, with the average usage of **Us**age microtheory facts being (expectedly) highest.

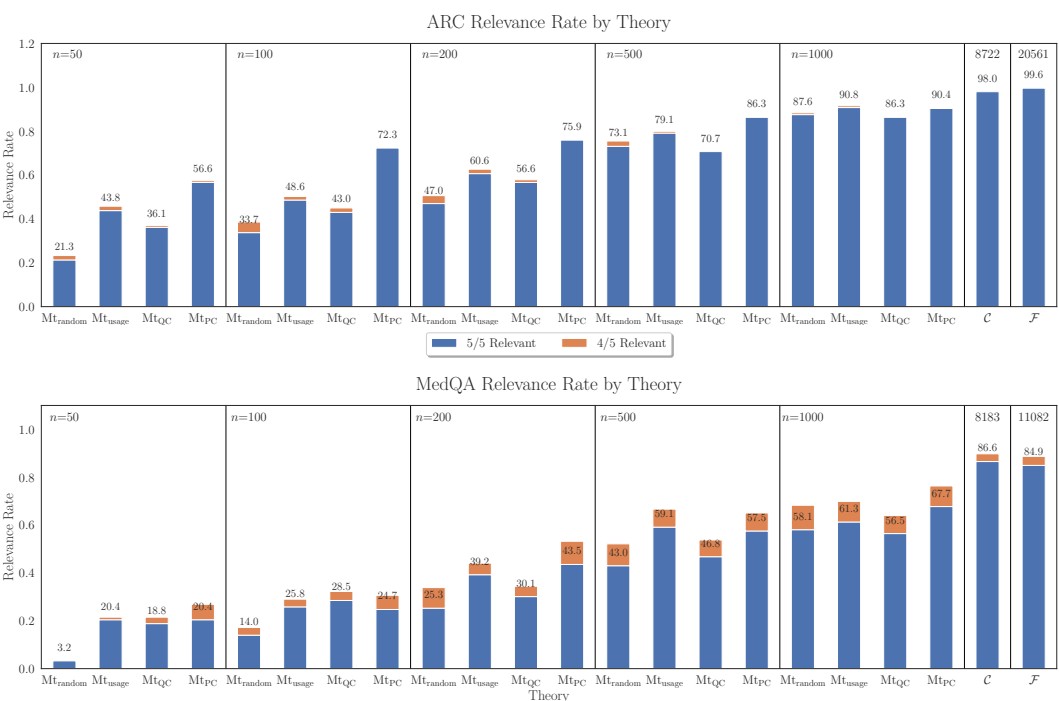

Figure 23: Rate (%) of microtheories containing at least one fact with 5/5 and 4/5 relevance as determined by GPT-4o. Numbers above each bar are the percentage of 5's.

## K AUTOMATIC RELEVANCE ANNOTATION DETAILS

The prompt used for scoring the relevance of facts to a given question is shown in Figure 26. The original rubric by Jansen et al. (2021) contains 4 ordinal scores; we added a 5th to differentiate between facts relevant to ruling out incorrect options (4) vs supporting correct options (5).

| # | Indicators | Exemplars |
|---|---|---|
| 5 | • [1] Statement/fact that includes relevant associations* and **IS** specific, comprehensive, or pathognomonic enough to provide a clear answer to a test question.
• [2] Statement/fact that provides **core** or **comprehensive** information and/or associations about a topic domain [B] that are **fundamental** to its understanding as a whole | [1] *Deletion of 11-p-15 can cause Beckwith-Wiedemann syndrome, which includes symptoms such as macroglossia, omphalocele, and hemihypertrophy*
[2] *Tachycardia (elevated heart rate) and tachypnea (elevated respiratory rate) are signs of cardiac or pulmonary distress.* |
| 4 | • [1] Statement/fact that includes relevant associations but **is NOT** specific, comprehensive, or pathognomonic enough to provide a clear answer to a test question
• [2] Statement/fact that provides **neither core nor comprehensive** information and/or associations about a topic domain that are relevant but **not fundamental** to its understanding as a whole
  ○ Without knowing this statement/fact, it **would** be difficult to correctly answer a question on this topic
• [3] Definition that **includes** relevant associations, context, and/or assumptions that are directly useful in answering a question correctly | [1] *In autosomal recessive inheritance, both parents must be carriers of the mutated gene for a child to inherit the disease*
[2] *Increased circulating estrogen can occur in men with liver disease, such as alcoholic cirrhosis*
[3] *Osteoporosis is a condition characterized by low bone mass and deterioration of bone tissue, leading to increased bone fragility and risk of fractures.* |
| 3 | • [1] Statement/fact that is accurate **and** provides context or information about a topic; **does not** provide information that would directly result in getting a question right.
  ○ Without knowing this statement/fact, it **would NOT** be difficult to correctly answer a question on this topic
• [2] Definition that **lacks** relevant associations, context, and/or assumptions that are directly useful in answering a question correctly
  ○ May include examples or additional details that do not directly facilitate a correct answer | [1] *Symptoms of drug-induced fever include fever and confusion*
[2] *A Physician Health Program is a program designed to help physicians with substance abuse or mental health issues* |
| 2 | • [1] Statement/fact that is **generic** or **common-sense;** information is widely known and/or lacks details or relevance about a given topic
• [2] Statement/fact that **rarely** helps in the specific context of medical exams, offering **little** value in the diagnostic and/or question-answering process
• Statement/fact that contains an **element or component** that is **factually untrue** or **inaccurate** | [1] *The spine is a part of the musculoskeletal system*
[2] *Two-dimensional gel electrophoresis is not typically used to diagnose genetic disorders*
[3] *Decreased activity of epithelial Na+ channels in principal cells can be caused by increased activity of luminal K+ channels in principal cells* |
| 1 | • [1] Statement/fact that is **useless**, **unclear**, and/or completely **irrelevant**
• [2] Statement/fact that **never** helps in the specific context of medical exams, offering **no** value in the diagnostic and/or question-answering process
• Statement/fact that is **entirely** factually untrue or inaccurate | [1] *The patient's BUN and creatinine levels are slightly elevated.*
[2] *The cause of erythema nodosum is often unknown, or may be related to certain systemic diseases.*
[3] *Bloody diarrhea can be a symptom of Giardia lamblia infection* |

*Relevant association: A significant connection or correlation between two or more related factors (e.g., diseases, symptoms, or risk factors) that directly impacts the understanding, pathophysiology, diagnosis, treatment, and/or prevention of a medical condition.

**Topic domains: A core subject, discipline, or system in medicine that encompasses many which many questions are derived from (e.g. cardiac system, renal system)

Figure 24: Rubric used to collect expert relevance ratings for medical microtheory facts (1 of 2)

- Would a medical student be in trouble if they didn't know this fact the morning of the exam? (yes = 5)
- Can you imagine up (or remember) a question for which this statement would directly answer the question? (yes = 5)
- Would a topic domain (e.g., "cardiovascular system") be hard to understand if you didn't know this fact? (yes = 5)
- Could the information feasibly be important to understanding the correct answer or any of the key terms being tested by questions or correct answers (e.g., definitions, etc.) but not the concept being tested itself? (yes = 4)
- Does the statement add correct and relevant supporting details that give context or depth to an explanation, facilitating understanding of the concept? (yes = 4)
- Is the statement relevant to a correct explanation of a correct answer but generally not required for a complete and correct explanation? (yes = 3)
- Is the statement a semantic claim that correctly identifies a term that might be related to questions or correct answers but not required to produce the correct answer or explain it? (yes = 3)
- Is the statement partially incorrect or contains an element of untruth? (yes = 2)
- Is the statement "common knowledge"? (yes = 2)
- Does the statement have information that is too general or is only marginally related to a given topic? (yes = 2)
- Is the statement completely irrelevant to a correct and complete explanation of any feasible correct answer? (yes = 1)
- Is the statement completely incorrect? (yes = 1)

Figure 25: Rubric used to collect expert relevance ratings for medical microtheory facts (2 of 2)

```
You are a scientific expert that helps other scientists determine whether various facts are relevant to
    explaining the correct answer to a question. Scientists give you a question and a list of facts that
    they believe are relevant to the question. Your job is to determine whether each fact is relevant to
    explaining the correct answer to the question using a provided rubric for scoring:

A fact has a score of 5 if it has the following indicators:
* core/critical fact/knowledge that explains why a correct answer is correct
* essential piece of information if explaining phenomenon in question to a toddler

A fact has a score of 4 if it has the following indicators:
* core/critical fact/knowledge that explains why an incorrect answer is incorrect

A fact has a score of 3 if it has the following indicators:
* moderately important fact necessary in a correct explanation but not being explicitly tested *underlying
    assumption necessary to understanding the terms used in the correct response and explanation
* defines terms used in the prompt and/or the correct answer
* semantic statements: correctly identify a term in the immediate question or correct answer
* common sense statement that is topically relevant but not necessary for differentiating correct from
    incorrect

A fact has a score of 2 if it has the following indicators:
* extra detail - explanations missing this fact are neither incorrect nor incomplete
* true relevant statement but not necessary for understanding phenomena in question

A fact has a score of 1 if it has the following indicators:
* irrelevant
* incorrect

You do NOT need to give a 5 for at least one fact. You should give a 5 only if the fact is absolutely
    necessary and core to correctly answer the immediate question (i.e. if someone does not know the fact,
    they cannot answer the question correctly).

Here are some of the questions that you ask yourself when considering a fact:
Does this statement directly answer the original question? (yes = 5)
* Is the information absolutely necessary to correctly answer the immediate question? (yes = 5)
* Is the information absolutely necessary to explain why another answer is incorrect for the immediate
    question? (yes = 4)
* Is the information important to understanding the correct answer or any of the key terms being tested by the
    immediate question or correct answer (e.g., definitions, etc.) but not the concept being tested itself?
    (yes = 3)
* Is the statement a semantic claim that correctly identifies a key term in the immediate question or correct
    answer in a way that is relevant to the phenomenon being tested? (yes = 3)
* Is the statement relevant to a correct explanation of the correct answer but not required for a complete and
    correct explanation? (yes = 2)
* Does the statement add correct and relevant supporting details that give context or depth to an explanation
    but are not required to make the explanation complete and correct? (yes = 2)
* Is the statement a semantic claim that correctly identifies a term that is related to the immediate question
    or correct answer but not required to produce the correct answer or explain it? (yes = 2)
* Is the statement completely irrelevant to a correct and complete explanation of the correct answer? (yes =
    1)
* Is the statement incorrect? (yes = 1)
```

Figure 26: Prompt used to score the question relevance of retrieved microtheory statements. Instructions are based on the rubric from Jansen et al. (2021).

