# OpenReview forum: "From Models to Microtheories: Distilling a Model's Topical Knowledge for Grounded Question-Answering"
_ICLR.cc/2025/Conference — ICLR 2025 Poster_

### Official Review · Reviewer_AcjP · 2024-10-28

**Soundness:** 3
**Presentation:** 3
**Contribution:** 3
**Rating:** 6
**Confidence:** 3

**Summary:**

This paper presents a method for transforming a model's latent topical knowledge into a microtheories, encapsulating reusable knowledge about a topic. The authors also conducted evaluations demonstrating that, when incorporated into the general corpus, microtheories significantly enhance the model's grounding ability in its answers.

**Strengths:**

1. The paper proposes an innovative approach to distill microtheories from a model, that aritculates the model's core knowledge about the dataset's topic.
2. The author conducted comprehensive evaluations, including human evaluation to show that the microtheories can both improve the model's ability to ground its answer and improve accuracy.

**Weaknesses:**

1. Dependence on Training Questions and Answer Options: The method relies heavily on a pre-existing set of training questions and answer options to generate microtheories such as ARC. This dependency restricts its applicability to new corpora that lack such structured data, making the approach challenging to transfer across domains or to corpora where questions and answers aren't predefined.

2. The paper could indeed improve by including more details on the filtering process in Section 3.2.2. Specifically, it doesn’t clarify how many microtheories were ultimately filtered out after applying the redundancy reduction techniques.

**Questions:**

1. In Figure 7, adding 10 random microtheories to the base corpus could make the performance drops a lot, could you please explain it?

2. See weakness 2

---

> ### Author Response · Authors · 2024-11-22
> **Author Response to Reviewer AcjP**
>
> Thank you for highlighting the innovativeness of our distillation method and the comprehensiveness of our evaluation!
>
> **1. Dependence on Training Questions and Answer Options:**
> > This dependency restricts its applicability to new corpora that lack such structured data, making the approach challenging to transfer across domains or to corpora where questions and answers aren't predefined.
>
> In fact, our method could also be easily applied to a corpus without predefined QA pairs, as numerous methods exist to generate QA pairs from sentences. In this case, those QA pairs would similarly drive the LLM to articulate underlying facts required to prove the answers, from which microtheories can be distilled. We will note this in the paper. It’s also worth noting that almost all learning systems require training QA data. In this sense, this work is no different from those learning systems (and perhaps even more flexible, as such training data can be automatically generated from a corpus).
>
> **2. Clarifications on filtering process**
> > The paper could indeed improve by including more details on the filtering process in Section 3.2.2. Specifically, it doesn’t clarify how many microtheories were ultimately filtered out after applying the redundancy reduction techniques.
>
> To clarify our terminology: in our paper, a “microtheory” is a collection of facts (not a single fact). Our filtering techniques take a large microtheory and filter it down to a smaller microtheory of the most powerful facts. Again, to clarify: a “10-Mt” is a 10-fact microtheory filtered from a larger pool, not 10 microtheories. The distillation process doesn’t filter out microtheories; rather it filters out candidate sentences for the microtheory that is being distilled.
>
> Regarding how many of those facts get filtered out: Figure 3 might help answer your question. The reduction techniques in S3.2.2 filter the raw fact pool $\mathcal{F}$ down to condensed pool $\mathcal{C}$ (see line 185). As we show in Figure 3, the size of the initial fact pools, $\mathcal{F}$ is 20K facts for ARC and 11K for MedQA, while the size of the condensed fact pools (from which a microtheory is then extracted), $\mathcal{C}$, is 8.7K for ARC and 8.1K for MedQA.  In other words, the redundancy reduction techniques filter around 55% of the ARC facts and 25% of the MedQA facts. We have added this to our updated pdf.
>
> Please let us know if there is anything else about the filtering process that is unclear to you.
>
> **3. Performance drop for smaller microtheories**
> > In Figure 7, adding 10 random microtheories to the base corpus could make the performance drops a lot, could you please explain it?
>
> Again just to be clear, Mt-10 is a single microtheory of 10 sentences, not 10 microtheories.
> Indeed if we just select 10 sentences at random for a microtheory, they are unlikely to be helpful for test questions (and may even confuse the entailment engine, which relies on the facts to guide its search) given the sentence pool contains irrelevant and idiosyncratic facts. Hence the low score for Rd (random). This is precisely the problem that the distillation process is intended to solve, namely distilling a small set of relevant, useful facts. The higher scores for the three ways of distilling facts (Us, QC, PC) suggest this has been achieved.
>
>
> We have added a line to S5.2 in our updated pdf explaining that this happened.

---

> > ### Comment · Reviewer_AcjP · 2024-11-26
> > **Follow up on Author Response**
> >
> > I appreciate the authors' detailed response and explanation, which have helped to clarify my concerns.
> >
> > Regarding the **Dependence on Training Questions and Answer Options**, I would like to further elaborate. Not all learning systems rely on explicit question-answer (QA) pairs, particularly in scenarios like pretraining large language models (LLMs) on diverse corpora. These pretraining corpora typically do not contain predefined QA pairs. Could you elaborate on how your method would extend to such situations?

---

> > > ### Author Response · Authors · 2024-11-28
> > > **Thank you for the follow-up!**
> > >
> > > Yes, indeed. Just to be clear (and as you note yourself in your Summary), our goal is not to learn from a pretraining corpora, but to materialize the results of that learning as a microtheory. In other words, a problem with LLMs is that it is unclear exactly *what* they have learned from their pretraining corpora - microtheories help reveal that. Furthermore, by tidying up that knowledge and feeding it back to a (fixed, off-the-shelf) LLM-based reasoning system, we show we can improve that system's accuracy and grounding (i.e., ability to construct  "proofs" from a corpus to an answer).
> > >
> > > So in one sense a microtheory *is* already derived from a pretraining corpus - it is a projection of a model's latent topical knowledge, learned from that corpus, into an explicit NL form. However, our approach does also require some seed input to define the topic of interest, and we use a set of QA pairs for this purpose. Note, though, that our method first converts the QA pairs (for correct answers) into declarative sentences (hypotheses, lines 142 and 170), so in reality it is sentences (hypotheses) that drive microtheory construction. Thus we could, in principle, use the sentences in a Wikipedia paragraph (say) as the seed input rather than QA pairs, and the output would be a microtheory listing the basic principles that the LLM believed that would help explain ("prove") those sentences. Hence our method could in principle be applied to such situations, although we did not try this - the reason we used train/test QA pairs in our paper was so that we had a clear way of evaluating the results. We hope this helps clarify what is going on!

---

> > > > ### Comment · Reviewer_AcjP · 2024-11-28
> > > >
> > > > I thank the reviewers for clarifying my concerns. I have raised my score.

---

### Official Review · Reviewer_4U6H · 2024-11-04

**Soundness:** 3
**Presentation:** 3
**Contribution:** 3
**Rating:** 8
**Confidence:** 3

**Summary:**

The paper proposes a framework of distilling a model’s knowledge of a certain scientific topic into an inspectable, verifiable form of microtheories, and shows that adding microtheories in the original corpus improves entailment engine performance.

**Strengths:**

1. The paper addresses an important problem of distilling a model’s knowledge of a certain topic into an inspectable, verifiable form
2. The method proposed by the paper can extract facts that are deemed as critically relevant by human experts
3. The distilled facts are helpful for entailment engines to perform QA tasks

**Weaknesses:**

1. The format of the input to the entailment engine is unclear. The engine takes in a mixture of original corpus such as Wikipedia and microtheories, but how these external information is structured is not clear.
2. The claim "Mt size correlates with QA accuracy" in Section 5.2 seems to be an overstatement, as the authors only show three bins (100, 500, 1000).
3. The fact that only adding more than 1000 facts can increase QA performance shows that current microtheory selection is unable to get the most relevant facts. In addition, 5000/8000 of facts for both datasets are never used in the proof. The authors should discuss 1) why it's happening and 2) the compute needed to retrieve a large enough fact corpus to improve QA performance, and the trade off between selection comprehensiveness and QA performance.
4. Although the paper claims to generate microtheory from both GPT4 and Mixtral to show that their method not only works on closed models, there's no comparison of performance and discussion on how different models may affect microtheory generation. The paper also does not explain why they use GPT4 for ARC and Mixtral for MedQA.

**Questions:**

1. I am confused about the "training questions", do you mean questions in the original training set of ARC and MedQA? Also, the word "training" appeared in this paper so many times in different contexts, "training questions" "training hypothesis" "training set" "training proofs"... but it seems like you are not training the QA module at all?

---

> ### Author Response · Authors · 2024-11-22
> **Author Response to Reviewer 4U6H**
>
> Thank you for your careful analysis of our experiments! We appreciate you highlighting the importance of our considered problem.
>
> **1. Entailment Engine I/O**
> > The format of the input to the entailment engine is unclear. The engine takes in a mixture of original corpus such as Wikipedia and microtheories, but how these external information is structured is not clear.
>
> The input to the entailment engine is a flat index of documents, where each document is typically one or several sentences, and a hypothesis to prove. The output is an entailment tree showing how the hypothesis follows from a subset of those documents via entailment (or null, if no tree can be found).
>
> In our case, we break the Wikipedia corpus into paragraph-sized chunks (“documents”) – specifically, the  2021-01-20 version of Wikipedia with standard 100-word chunks. The microtheory is structured similarly, as a set of individual fact statements (each sentence is a “document”). Until they are used to compose new entailment trees at test time, the individual statements in the microtheory are not tied together in any way; it is just a large list of statements.
>
> When the search is given both the microtheory and Wikipedia, we concatenate the results in that order– microtheory statements first. The search then uses the concatenated set for entailment lookups and hypothesis decomposition. We’ve updated the appendix to include this information.
>
> **2. Fact Usage Rates**
> > 5000/8000 of facts for both datasets are never used in the proof.
>
> To clarify: In the raw fact pool, *before* we start distilling any microtheory, only 3000 of 8000 (i.e., about 40%) statements are used in the training proofs. This highlights the problem that our method aims to solve: simply gathering relevant facts produces a significant amount of unnecessary, duplicate, and/or unhelpful information. In contrast, essentially all (~100%) sentences in the distilled microtheory play a role in proving the training questions by virtue of the microtheory construction process, reflecting the Mt’s concentration of core, important facts, as well as helping ground test questions more effectively. This is a key contribution of the work.
>
> **3. Whether the findings support that the microtheory selection finds the most relevant facts**
>
> > The fact that only adding more than 1000 facts can increase QA performance shows that current microtheory selection is unable to get the most relevant facts.
>
> Note that our primary goal is to distill core, relevant facts that can help ground answers in multiple settings while not hurting QA performance (and, of course, any performance gains are an added bonus). We do see evidence that this distillation goal has been achieved even for the small Mts in multiple experiments. For example, for the 100-Mts and 500-Mts, Figure 6 (S5.1) shows that they provide more hypothesis grounding over the random baselines of the same size. Figure 8 (S5.3.1) shows that they are more relevant under expert human evaluation. We've also updated Figure 9 (S5.3.2.) to show the difference in "at least one relevant fact" rate between a distillation method and the random baseline. This suggests the more relevant facts have been distilled for the smaller sizes. Note that there is no guarantee that a good Mt will improve QA, as the entailment engine may then just ground to the large corpus instead (Wikipedia). But this does nothing to distill the LLM’s core knowledge about the topic, which is the goal of this work.
>
> **4. Discussion of required compute**
> > the compute needed to retrieve a large enough fact corpus to improve QA performance, and the trade off between selection comprehensiveness and QA performance.
>
> Yes, this is important. This is what motivated our new metric, $p$-relevance, in Section 6.
>
> For a fact corpus to be relevant to a given question, it should contain at least one fact that is critical to answering the question correctly. We found in S5.3.2 that the rate at which the microtheory corpus contains at least one critical fact per question is proportional to the raw size of the corpus, hence there is a clear trade-off between storage and coverage. You’re right that compute is another critical consideration, since creating the distilled microtheories requires running the entailment engine on each training question, which can be costly. We’ve added language to section 6 discussing this.
>
> We also found that even the raw fact pool F doesn’t contain at least one relevant fact for 100% of the questions. The $p$-relevance metric introduced in S6 measures how many training questions we’d need to extract facts for - hence the additional compute needed - in order to be relevant to questions some $p\%$ of the time, hence helping characterize the compute required.

---

> ### Author Response · Authors · 2024-11-22
> **Author Response to Reviewer 4U6H (continued)**
>
> **5. Comparison of multiple models on the same dataset**
> > Although the paper claims to generate microtheory from both GPT4 and Mixtral to show that their method not only works on closed models, there's no comparison of performance and discussion on how different models may affect microtheory generation. The paper also does not explain why they use GPT4 for ARC and Mixtral for MedQA.
>
> Thank you for pointing this out. We’ve updated the paper to include using GPT-4 for MedQA so that the results between models are directly comparable. We find building and using Mts with Mixtral to be as effective as using GPT4, in terms of grounding and relevance, showing our method is not restricted to GPT4 only. We also re-ran our human evaluation using the GPT4-generated microtheories.
>
> **6. Clarifications of training scenario**
> > I am confused about the "training questions", do you mean questions in the original training set of ARC and MedQA?
>
> Almost, except we created our own train/test split for topical subsets of ARC and MedQA (see lines 258–264 and Appendix B). This was because we wanted train/test questions about the same topic (e.g. all the ARC questions that target knowledge of mechanical physics), so that the test set is a proper evaluation of whether the Mt has captured knowledge about that topic.
>
> > Also, the word "training" appeared in this paper so many times in different contexts, "training questions" "training hypothesis" "training set" "training proofs"... but it seems like you are not training the QA module at all?
>
> Correct, we are not training the QA module directly, e.g. modifying parameters of a neural net. However, there is some learning going on: The performance of the (fixed) QA module depends on the input knowledge (e.g., Wikipedia, a microtheory) that it takes as input, and hence by improving those resources (i.e., building a good microtheory), the performance of the overall system improves (grounding, QA). Hence, the train/test paradigm: training is used to help build the knowledge resource, and a test set is used to evaluate how successful that has been.

---

> > ### Comment · Reviewer_4U6H · 2024-11-25
> > **Follow up on Author Response**
> >
> > I thank authors for their additional experiments and explanations. I have two follow up questions:
> >
> > 1. Section 5.3 uses "relevance" as a metric to evaluate how the microtheory building process is indeed extracting the more important, central facts about the topic. This is a mismatch: relevant information may not be selected by the entailment engine or really used by the QA model. I think the authors need to add another experiment on how many facts in the microtheory is used by the entailment engine, or use causal metrics like influence function to show true causality that certain facts are indeed used by the QA model in answers.
> >
> > 2. Which QA models are you using in Section 5.2?

---

> > > ### Author Response · Authors · 2024-11-26
> > > **Thank you for the follow-up**
> > >
> > > **1. Additional Usage Metrics**
> > > We have added a new set of usage metric results to the paper, counting (1) how many facts in the microtheory are used by the entailment engine as a fraction of the entire microtheory and (2) the average number of questions per fact.  They are in "Appendix H: Additional Fact Usage Metrics."  In both cases, the selection methods (usage-optimized, PC, QC) substantially outperform both the random baseline and the full fact pool $\mathcal{F}$, much of which is never used at test time. The usage-optimized Mt expectedly performs the best since it was created by sorting the facts by usage count on training questions and taking the top $n$.
> > >
> > > It's important to note that these usage statistics don't capture functional redundancies: if two facts serve the same role in entailment, they will be frequently used at the same rate. This will show up as scoring high under the "fraction microtheory used" metric, even though it is undesirable to us because keeping both of them in the microtheory is risking keeping out a less frequently used, but still important to keep, fact.  We discuss this in section 3.2.3:
> > >
> > > > After removing duplicates and entailments, we still see facts that serve the same _functional_ roles in many questions but imply slightly different things, e.g. for physics:
> > >
> > > > (A) The force required to cause a given acceleration is determined by the mass of an object
> > >
> > > > (B) The force acting on an object equals mass times acceleration
> > >
> > > > While this might be acceptable for a knowledge store with infinite storage, we instead seek a more concise representation without redundancies. [...] we would not want to retain both (A) and (B) in the $n$-Mt if it meant discarding some (C) that is important to other questions.
> > >
> > > To illustrate this problem: if we took the top-1 most frequent fact and rephrased it 100 times to create a microtheory, this microtheory would have a high score under the usage metrics, but it wouldn't be a useful microtheory for most of the questions at test time.
> > >
> > > This is the reason we focused on test hypothesis grounding rate in the paper (Section 5.1 and Figure 6). This measures the fraction of test proofs grounded to the microtheory. But you are correct that these extra usage statistics help give a broader picture of how the different MTs behave. We appreciate the feedback and think the paper will benefit from these additional results.
> > >
> > > **2. Which QA models?**
> > > We are using the entailment engine for QA. We have the engine search for entailment proofs for each answer option, then take as its answer the option with the highest scoring proof. The engine gets no credit if it fails to find a proof for any option.
> > >
> > > Specifically, we use TreeWise [[1](https://aclanthology.org/2024.emnlp-main.531/)] as the entailment engine, which uses GPT-3.5-turbo to perform the search.

---

> > > > ### Comment · Reviewer_4U6H · 2024-11-26
> > > > **Thank you for your response**
> > > >
> > > > I thank the authors for the prompt response. I am raising my score to 8.

---

### Official Review · Reviewer_qyBe · 2024-11-04

**Soundness:** 3
**Presentation:** 3
**Contribution:** 2
**Rating:** 6
**Confidence:** 3

**Summary:**

This paper presents a novel approach to enhancing question-answering systems by distilling a model's topical knowledge into what the authors term "microtheories." The method involves generating a set of simple, logical facts about the world, which are used to construct entailment chains that support the answering of complex queries. The paper emphasizes the importance of creating relevant and context-specific facts, which can be either generic statements about the world or specific assertions that aid in proving a particular query. The authors introduce a system that generates query statements for each possible answer to a question, ensuring these queries align closely with the answer choices. The effectiveness of this approach is evaluated through expert-annotated task relevance scores, demonstrating that microtheory-based facts are more relevant than random or usage-based facts, particularly when constrained by a fact budget. This work aims to improve the reasoning capabilities of AI systems by providing a structured framework for knowledge representation and retrieval, ultimately leading to more accurate and grounded question answering.

**Strengths:**

1. The paper presents a novel concept of generating microtheories from models. By utilizing advanced models like GPT-4 and Mixtral-8x22B-Instruct-v0.1, the authors demonstrate a robust approach that is not dependent on a single closed model. This flexibility and innovation in methodology can potentially lead to more accurate and contextually relevant answers in reasoning tasks.

2. Comprehensive evaluation. This thorough evaluation process ensures that the proposed method is not only theoretically sound but also practically effective on three domains. The focus on optimizing microtheory contents given a size budget highlights the paper's attention to practical constraints and efficiency, making it a valuable contribution to the field.

**Weaknesses:**

- Line 74: The *entailment* in NLP typically is typically referred to as textual entailment, which surely has a formal [definition](https://en.wikipedia.org/wiki/Entailment_(linguistics)).
- Line 160-161, the method that you extract is using LLM prompting. The problem is, what's the inherent difference of your method compared to chain/tree/graph-of-thought, since you are both distilling knowledge from LLM as intermediate steps. I get it that you used the tricks like entailment consolidation to filter some meaningful microtheories, but there are many existing works focusing on filtering intermediate thoughts or supporting facts to facilitate CoT, which limits the novelty of your work.

**Questions:**

1. How is the *grounding* metric in section 5.1 calculated in detail?
2. How do you detach the improvements brought by *microtheory* and that possibly brought by seemingly useful LLM-generated chain-of-thought?

---

> ### Author Response · Authors · 2024-11-22
> **Author Response to Reviewer qyBe**
>
> Thank you for your careful review, highlighting the novelty of our generation method and the comprehensiveness of our evaluation!
>
> **1. Differences with Chain-of-Thought**
>
> > what's the inherent difference of your method compared to chain/tree/graph-of-thought, since you are both distilling knowledge from LLM as intermediate steps
>
> > How do you detach the improvements brought by microtheory and that possibly brought by seemingly useful LLM-generated chain-of-thought?
>
> The goals of chain-of-thought and of microtheories are quite different. Chain/tree/graph-of-thought methods show transient fragments of knowledge for individual questions, but they don’t reveal the "big picture": a single, concise store of discrete knowledge that is generalizable as a source of argument grounding for multiple questions in a topic or dataset (see Figure 1).  Our goal is to attempt to build this store: e.g., what is (say) GPT-4’s core knowledge about physics or medicine? This is a non-trivial task since just unioning many machine-generated sentences together results in duplicates, near-duplicates, over-generalizations, under-generalizations, unhelpful idiosyncratic facts, etc.
>
> So, the heart of our work is a careful distillation process to condense these into a parsimonious, most-useful theory (which you have noted in your Strength #2). This is the first time such theories have been generated from LMs, allowing people to see the LM's underlying "picture" and improving grounding and accuracy.To our knowledge, this is the first system to achieve this.
>
> In addition, the context of our work is to produce well-defined entailment arguments (“proofs”) that go from facts in an inspectable corpus (e.g., a microtheory, Wikipedia) to an answer, i.e., *grounded* proofs. This is different from chain/tree/graph of thought, which generates informal arguments supporting an answer and which are ungrounded (i.e., not connected to an external corpus).
>
>
> **2. Regarding a formal definition of textual entailment**
> > Line 74: The entailment in NLP typically is typically referred to as textual entailment, which surely has a formal definition.
>
> In fact, entailment doesn’t have a fully formal definition in the literature due to its reference to human understanding. The standard definition used by the NLP community, which we adopt here also, is:
>
> “We say that T entails H if the meaning of H can be inferred from the meaning of T, as would typically be interpreted by people. This somewhat informal definition is based on (and assumes) common human understanding of language as well as common background knowledge” – [Dagan et al (2005)](https://api.semanticscholar.org/CorpusID:8587959)
>
> Despite this semi-formative approach, recent works have produced reasonably reliable entailment engines from large amounts of training data, including the off-the-shelf one that we used in this work.
>
> **3. Grounding Metric**
> > How is the grounding metric in section 5.1 calculated in detail?
>
> We take a dataset of questions and measure the fraction of correct answers that can be grounded, i.e., “proved” via entailment from sentences in a discrete, authoritative corpus. We have added a clarification for this in S5.1 of the new pdf.
>
> The engine converts each correct answer to a hypothesis and runs an entailment tree search that returns {grounded} or {not grounded}. In the {grounded} case, it also returns one or more entailment trees. The leaves of the trees are either facts in the base corpus, or facts from the microtheory. The solid-colored bars in Figure 6 show the fraction of all tree leaves that came from the microtheory, while the striped-colored bars show the fraction that came from the base corpus.
>
> If the prover finds multiple trees, we count the leaves in the one with the highest fraction of microtheory leaves (footnote 8).

---

> > ### Comment · Reviewer_qyBe · 2024-11-26
> > **Response**
> >
> > Thanks for the clarification and I find the paper quite clear now. I'll raise the score to 6 for the soundness. However, the topic of the paper is of a relatively scope so 6 should be a reasonable rating.

---

> > > ### Author Response · Authors · 2024-11-26
> > > **Thank you!**
> > >
> > > Thank you for stating that you'll raise your score for our paper! We look forward to seeing the review updated with the score adjustment.

---

### Author Response · Authors · 2024-11-22
**General Response + PDF Updates**

We thank all reviewers for taking the time to carefully review our paper's merits and provide constructive feedback. We have responded to reviewers' concerns on a case-by-case basis, and have made the following updates to the submission based on the discussion:

* **[R.qyBe Q1]** Added a line of clarification for the grounding metric in S5.1.
* **[R.4U6H W1]** Added clarifying details to _Appendix F: Retrieval Implementation Details_ about the format of the retrieval indices given as input to the entailment engine.
* **[R.4U6H W3]** Modified Figure 9 to clarify the difference between the distilled microtheory and random baseline under our automated "at least one relevant fact" relevance rate metric.
* **[R.4U6H W3]** Added discussion to S6 about the scaling compute required to create a microtheory for $n$ training questions.
* **[R.4U6H W4]** Added results across S5 for GPT-4 on the MedQA dataset so that results are comparable across datasets and models.
* **[R.AcjP W1]** Added an explanation for the QA performance drop using the smaller random baseline Mts


We look forward to further discussion with reviewers during the remainder of the discussion period to iron out any further concerns they might have.

---

### Author Response · Authors · 2024-11-25
**Nearing the end of PDF upload period**

We once again thank all reviewers for their feedback so far.

While the phase has been extended 6 days for posting messages, the PDF upload deadline remains tomorrow. We would appreciate it if any reviewers would comment as to whether we have addressed their concerns, or if there are remaining issues that we can address through further changes to the pdf submission.

Thanks!

---

### Meta-Review · Area_Chair_o9c2 · 2024-12-20

**Metareview:**

The paper proposes to distill an LLM's knowledge about a topic into "microtheories", which can be combined to form entailment chains to answer complex queries. Overall, the reviewers agree that the idea is quite promising. There are some concerns about how this method extends to settings where a training corpus of QA pairs is not available, or when there is no overlap in topics between the train/test splits.

**Additional Comments On Reviewer Discussion:**

Reviewer qyBe raised questions about the approach's difference from CoT reasoning, which the authors sufficiently address in their response. Reviewer 4U6H raised some questions about how many facts does the micro theory need to be comprised of, and questions whether meaningful improvements in QA are seen without generating 1000s of facts. The authors argue that the goal of the work is to distill core facts about a topic to provide grounding evidence, and not necessarily improve QA performance. There are also some concerns about the clarity of the paper.

---

### Decision · Program_Chairs · 2025-01-22

Accept (Poster)